# Genome-wide CRISPR screens identify GATA6 as a proviral host factor for SARS-CoV-2 via modulation of ACE2

Ma'ayan Israeli[1,5], Yaara Finkel [2,5], Yfat Yahalom-Ronen[3], Nir Paran [3], Theodor Chitlaru[1], Ofir Israeli[1], Inbar Cohen-Gihon[1], Moshe Aftalion[1], Reut Falach[1], Shahar Rotem[1], Uri Elia[1], Ital Nemet[4], Limor Kliker[4], Michal Mandelboim[4], Adi Beth-Din[1], Tomer Israely [3], Ofer Cohen[1], Noam Stern-Ginossar [2] & Adi Bercovich-Kinori [1✉]

The global spread of SARS-CoV-2 led to major economic and health challenges worldwide. Revealing host genes essential for infection by multiple variants of SARS-CoV-2 can provide insights into the virus pathogenesis, and facilitate the development of novel therapeutics. Here, employing a genome-scale CRISPR screen, we provide a comprehensive data-set of cellular factors that are exploited by wild type SARS-CoV-2 as well as two additional recently emerged variants of concerns (VOCs), Alpha and Beta. We identified several host factors critical for SARS-CoV-2 infection, including various components belonging to the Clathrin-dependent transport pathway, ubiquitination, Heparan sulfate biogenesis and host phos-phatidylglycerol biosynthesis. Comparative analysis of the different VOCs revealed the host factors KREMEN2 and SETDB1 as potential unique candidates required only to the Alpha variant. Furthermore, the analysis identified GATA6, a zinc finger transcription factor, as an essential proviral gene for all variants inspected. We show that GATA6 directly regulates ACE2 transcription and accordingly, is critical for SARS-CoV-2 cell entry. Analysis of clinical samples collected from SARS-CoV-2 infected individuals shows elevated levels of GATA6, suggesting a role in COVID-19 pathogenesis. Finally, pharmacological inhibition of GATA6 resulted in down-modulation of ACE2 and inhibition of viral infectivity. Overall, we show GATA6 may represent a target for the development of anti-SARS-CoV-2 therapeutic strategies and reaffirm the value of the CRISPR loss-of-function screens in providing a list of potential new targets for therapeutic interventions.

---

[1] Department of Biochemistry and Molecular Genetics, Israel Institute for Biological Research, Ness Ziona, Israel. [2] Department of Molecular Genetics, Weizmann Institute of Science, Rehovot, Israel. [3] Department of Infectious Diseases, Israel Institute for Biological Research, Ness Ziona, Israel. [4] Central Virology Laboratory, Public Health Services, Ministry of Health and Sheba Medical Center, Tel Hashomer, Ramat Gan, Israel. [5] These authors contributed equally: Ma'ayan Israeli, Yaara Finkel. ✉email: adiki@iibr.gov.il

Severe Acute Respiratory Syndrome Coronavirus 2 (SARS-CoV-2) is a recently emerged type of respiratory syndrome-related coronaviruses, the cause of Coronavirus Disease 2019 (COVID-19), responsible for the most challenging pandemic in this century. Despite an unprecedented worldwide research effort that resulted in the rapid development of a variety of vaccines against SARS-CoV-2, the pandemic remains uncontrolled in many countries and continues to take a devastating toll on both human health and global economic activity. The emergence of SARS-CoV-2 variants with enhanced transmissibility and pathogenesis, or which may evade pre and post-exposure countermeasures, presents further challenges in the continuous struggle against the virus and the tragic consequences of the pandemic.

Coronaviruses are enveloped, positive-sense single-stranded RNA viruses with a genome of ~30 kb in length[1]. The initial steps of SARS-CoV-2 infection involve the specific binding of the coronavirus spike protein (S) to the cellular entry receptor angiotensin-converting enzyme 2 (ACE2)[2–5]. The engagement of the receptor and the subsequent membrane fusion commences the process of viral entry into the host cell by endocytosis allowing the virion to release the genetic material into the cytoplasm. In the host-cell cytoplasm, the viral RNA is translated into continuous polypeptides that are cleaved into 16 nonstructural proteins, which then facilitate the transcription of sub-genomic RNA that is translated into structural and accessory proteins. These viral proteins rearrange host membranes to form the endoplasmic reticulum-localized viral replication complex (RC) in which the viral genomic RNA is replicated by the viral RNA-dependent RNA polymerase (RdRP) complex. The assembled components further undergo maturation in the Golgi apparatus to form the mature virion that is released outside the host cell by exocytosis[1,6–8]. Every step of the viral life cycle, from entry to budding, is orchestrated through interactions with host cellular proteins. For example, cellular proteases, such as TMPRSS2, Cathepsin L, and Furin, are exploited for the cleavage of the viral spike protein of SARS-CoV-2, mediating efficient membrane fusion with the infected cell[9–13]. Identification of additional key host proteins involved in the various steps of the infection is essential for the development of countermeasures and represents the objective of intense efforts. Several proteomics studies addressed the SARS-CoV-2-host interactions by utilizing immunoprecipitation mass spectrometry (IP-MS) of affinity-tagged viral proteins and proximity labeling of the virus replicase complex[14–17]. The proteomics approach revealed comprehensive interactomes and physical contacts between many viral and cellular proteins and highlighted potential targets for drug repurposing. Yet, this approach could not provide direct information pertaining to the essentiality of these host components for enabling the SARS-CoV-2 life cycle. Another strategy to explore SARS-CoV-2-host interactions is to globally disrupt individual genes of the entire host genome and screen for those whose disruption resulted in resistance to viral infection and consequently to host-cell survival. Several reports documented genome-wide CRISPR-Cas9-mediated loss-of-function screens and identification of host factors that are functionally required for SARS-CoV-2 infection in various cell types[10,18–25]. These studies identified genes, metabolic, and signaling networks that previously have not been considered as potential therapeutic targets for SARS-CoV-2, such as the cholesterol synthesis pathways[19].

As of today, following a year-long period of combating Covid-19 essentially by medical care and social distancing, the contribution of many effective vaccines started to materialize[26,27]. Covid-19 vaccines exhibited as high as 95% efficacy in preventing clinical cases and 100% efficacy in preventing severe disease with the original virus[28,29]. The average mutation rate of SARS-CoV-2 remains low and steady. However, the continuing global spread of the SARS-CoV-2, together with selective pressure for immune escape, led to the emergence of new SARS-CoV-2 variants raising concerns regarding their relative ability to escape from natural and vaccine-induced immunity. Four novel variants that generated different patterns in the pandemic expansion, are of considerable public-health concern, as concluded by the WHO organization following judicious monitoring and classification of the new lineages (hence eloquently coined VOC, acronym for variants of concern): (i) Alpha (also known as B.1.1.7, VOC-202012/01- Variant of Concern, year 2020, month 12, variant 01) emerged in the UK. (ii) Beta variant (also known as B.1.351 variant, 501Y.V2), emerged in South Africa. (iii) Gamma, (also known as the P.1 lineage, 501Y.V3 variant), first identified in Brazil (iv) Delta (also known as B.1.617.2. variant) originally surfaced in India[30–33]. The Alpha lineage has a total of 17 non-synonymous mutations relative to the original Wuhan strain, of which 7 replacements and 2 deletions reside in the spike protein. The Alpha variant was shown to be significantly more infectious than other lineages (increasing the effective reproduction number, $R_0$, by a factor of 1.35) which resulted in its rapid global expansion. As of May 2021, the variant has been detected in over 100 countries, with a worldwide daily prevalence of over 75%[34]. The Beta variant emerged independently of the Alpha variant but shares with it mutations at several loci. The significant anxiety caused by this variant, owing to the co-occurrence of the N501Y mutation in the spike protein receptor-binding domain (RBD, also found in the Alpha variant) with additional mutations K417N/T and E484K. Viral variants with the triple combination of mutations exhibit reduced susceptibility to vaccine-induced and convalescent sera[35–41]. Fortunately, the Beta variant is less abundant with a reported worldwide daily prevalence of 1%[42,43]. In addition to the assessment of full immunological and clinical implications of the SARS-CoV-2 new variants, there is an urgent need for a more profound understanding of their biology, and for integrating this information towards the development of therapeutic approaches targeting host elements essential to all variants.

In this study we have performed a genome-wide CRISPR loss-of-function screen in a lung-derived human cell line infected with SARS-CoV-2. In an attempt to probe shared and differential host factors that may be essential for the virus in the course of infection, the genomic approach was expanded to the original WT-SARS-CoV-2 and two additional variants, Alpha and Beta. This strategy led to the discovery of known and novel SARS-CoV-2-host interactions, enabled the identification of both common factors as well as those specific for a particular variant, and revealed a pivotal role of the host pleiotropic regulator GATA6, which may represent a promising target for therapy.

## Results and discussion

**Genome-wide CRISPR screens for the identification of host factors essential for SARS-CoV-2 infection.** To identify host factors essential for viral infection and/or for cell survival in response to human SARS-CoV-2, a CRISPR-based genome-wide gene-knockout screen was performed using the human lung epithelial cell line Calu-3, which is highly permissive to SARS-CoV-2 infection due to elevated levels of endogenous ACE2 expression. The library employed for the screen was the Brunello CRISPR library composed of 76,441 targeting single guide RNAs (sgRNAs) with an average of four sgRNAs per gene and 1000 non-targeting control sgRNAs[44]. For the identification of host factors exhibiting either broad or restricted essentiality for the different variants of SARS-CoV-2, Calu-3 cells transduced with the CRISPR library, were infected separately with each of the

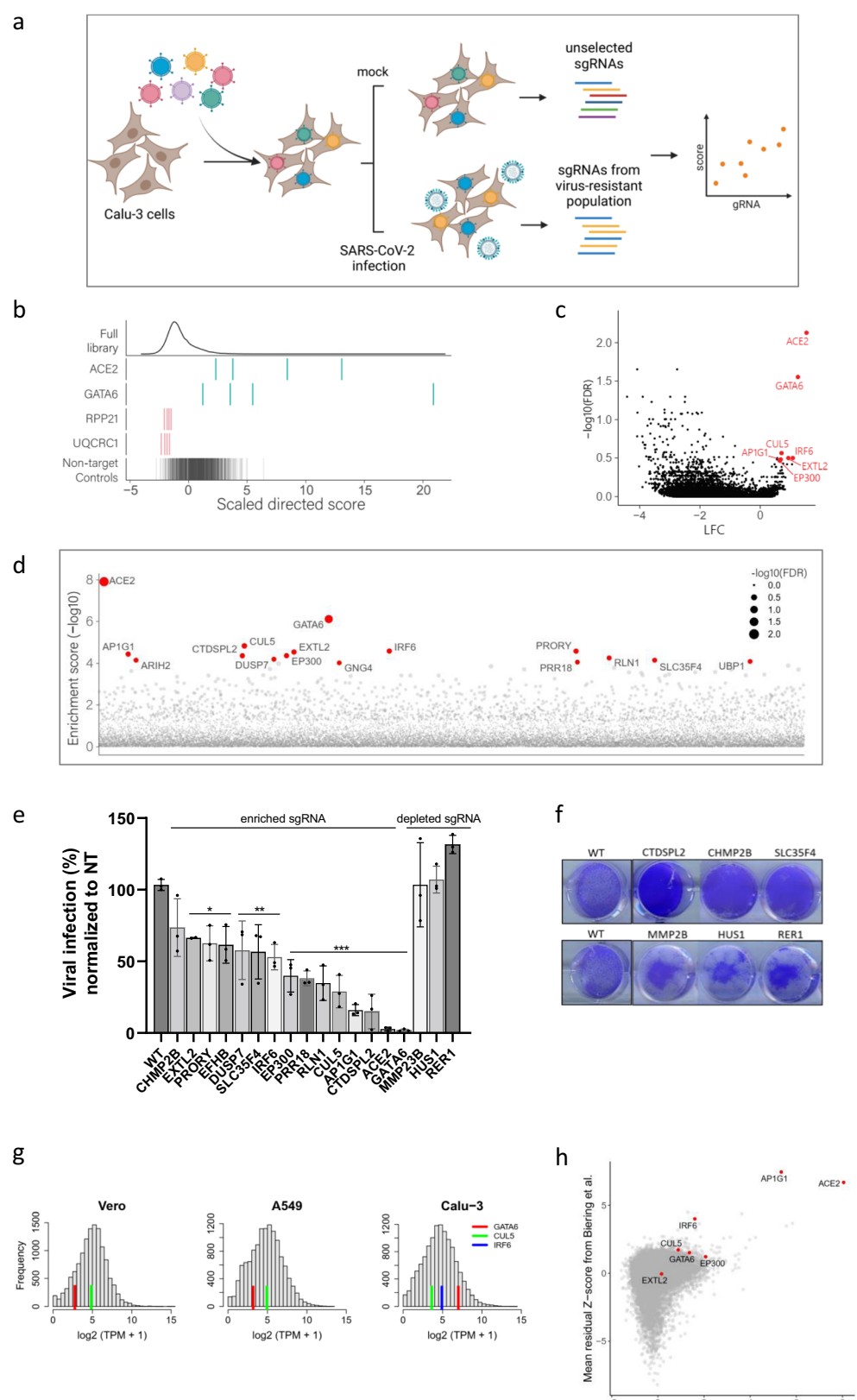

SARS-CoV-2 variants of concern (VOCs): SARS-CoV-2 B.1.1.7 (Alpha), SARS-CoV-2 B.1.351 (Beta), and the original wild-type (WT) SARS-CoV-2. Genomic DNA was harvested from surviving cells 7–9 days post-infection, and sgRNAs abundance was determined by sequencing (Fig.1a). In the initial analysis, to identify cellular functions important for infection of

Calu-3 cells with SARS-CoV-2, irrespective of the variant, the sgRNA distributions measured in the 3 screens were collectively compared to the control non-infected cells. The analysis identified numerous genes that upon disruption, conferred resistance to death by viral infection (referred to as proviral) or sensitization to viral infection (referred to as antiviral) (Supplementary Data 1).

**Fig. 1 CRISPR genome-wide screens in human Calu-3 cells identify host factors important for infection by wild-type SARS-CoV-2 and variants of concern. a** Schematic of genome-wide CRISPR screening workflow for the identification of SARS-CoV-2-host factors. Calu-3 cells were transduced with the Brunello CRISPR library, selected with puromycin and infected with WT-SARS-CoV-2, Alpha or Beta VOCs. Surviving cells were harvested 7–9 days post-infection (dpi). The abundance of each sgRNA in the mock controls and selected population was determined by high-throughput sequencing and a gene enrichment analysis was performed. The figure was created with BioRender.com. **b** Performance in the screens of example sgRNAs. The density across the screens is plotted for the full library (top) and for the 4 sgRNAs targeting top resistance hits, top sensitization hits and the non-targeting control sgRNA. **c** Volcano plot showing top genes conferring resistance and sensitivity to SARS-CoV-2. **d** Gene enrichment score for CRISPR screens of WT-SARS-CoV-2 and VOCs infection. Enrichment scores were determined by MaGECK. **e** Real-time PCR quantification of WT-SARS-CoV-2 levels in wild-type and CRISPR-edited Calu-3 cells. A non-targeting (NT) sgRNA was used as control. Cells were infected using MOI = 0.002 for 48 h. Data were analyzed from three biologically independent experiments by one-way ANOVA with two-sided Tukey's multiple comparison test. Shown are means ± SD. *$p < 0.05$; **$p < 0.005$; ***$p < 0.0001$. **f** Cell viability assay of wild-type and individual CRISPR-edited Calu-3 cells infected with SARS-CoV-2 at MOI of 0.002 for 48 h, and stained with crystal violet. One of three repetitions is shown. **g** Histograms of mRNA levels in Calu-3, Vero-E6, and A549 cells[66–68], normalized to transcripts per million (TPM). Red, green, and blue lines represent the levels of GATA6, CUL5, and IRF6, respectively, in each dataset. IRF6 is below threshold in A549 and Vero-E6 cells. **h** Comparative analysis of previously reported SARS-CoV-2 knockout screens performed in the Calu-3 cell line[21,62]. Top proviral candidates identified in our screens are displayed. Source data are provided as a Source Data file.

The sgRNA score-distribution of the full library, non-targeting control sgRNAs, and that of top enriched and depleted- sgRNAs hits demonstrated the high technical quality of the screens (Fig. 1b and Supplementary Data 2).

As expected, the proviral gene, exhibiting the highest enrichment score was *ACE2*, the receptor serving as the portal for SARS-CoV-2 cell entry (log fold-change (LFC) = 1.534, false discovery rate (FDR) = 0.007). The transcription factor GATA6 (GATA Binding Protein 6) scored as the second-strongest proviral hit (LFC = 1.24, FDR = 0.028, Fig. 1c, d). GATA6 is a DNA-binding protein belonging to a family of zinc-finger transcription factors that participate in the regulation of cellular differentiation during vertebrate development and contribute to immune regulation through NF-kB and various other pathways. Interestingly, the transcription level of GATA6 was shown to be elevated in the lung transcriptome of COVID-19 patients strongly implying that GATA6 takes part in the viral-host crosstalk[45]. Other top proviral hits included *CUL5, IRF6, AP1G1, EP300,* and *EXTL2* (Fig. 1c, d). The *CUL5* gene encodes Cullin5, a core component of multiple SCF-like ECS (Elongin-Cullin 2/5-SOCS-box protein) E3 ubiquitin-protein ligase complexes, which mediate the ubiquitination and subsequent proteasomal degradation of target proteins. While ubiquitination of viral proteins can be regarded as a host-defense mechanism for destroying the incoming pathogen, viruses have adapted to exploit this cellular process to enhance various steps of the replication cycle and increase pathogenesis[46]. Indeed, in addition to the highly modified ubiquitin-proteome of SARS-CoV-2-infected host cells, ubiquitination modifications were observed also on SARS-CoV-2 proteins, and these modifications were reported to inhibit the host innate immune response[46–48]. *IRF6* gene is a transcriptional activator with a well-established role in the development of the epidermis. A recent study suggests a role for IRF6 in the modulation of the NFk-B-pathway that influences cell survival by regulating host responses during viral infection, such as cytokine production[49]. *AP1G1* (Adaptor Related Protein Complex 1 Subunit Gamma 1) encodes a component of clathrin-coated vesicles that were shown to mediate endocytosis and intracellular trafficking of SARS-CoV-2[50,51]. Of note, the clathrin-coating complex, to which AP1G1 belongs, was recently suggested to be important for SARS-CoV-2 infectivity in Calu-3 cells but not in other cell lines[21]. *EP300* encodes a histone acetyltransferase that is recruited by SMAD complexes to activate target genes in response to TGF-β receptor signaling[52]. Interestingly, in a recent search for functional modifiers of ACE2 surface abundance, EP300 was found to regulate ACE2 transcription and hence to influence cellular susceptibility to SARS-CoV-2 infection[53]. *EXTL2* encodes a glycosyltransferase involved in the chain elongation step of

heparan sulfate (HS) biosynthesis[54,55]. Several heparan sulfate biosynthetic genes were also markedly enriched in other CRISPR-knockout SARS-CoV-2 screens conducted in Huh7.5.1 cells[22,23] consistent with a recent report that SARS-CoV-2 infection is co-dependent on heparan sulfate and ACE2[56]. Since HS is ubiquitously expressed on the surfaces and in the extracellular matrix of all cell types, EXTL2 may play a central proviral role through heparan sulfate production that may determine SARS-CoV-2 tropism.

In addition to sgRNAs that were enriched following infection, the screen enabled the identification of sgRNAs that were depleted in virus-infected cultures suggesting the genes targeted by these sgRNAs may have an antiviral function. However, gene disruption by itself may affect cell viability resulting in non-specific susceptibility to infection. To further evaluate whether the significant depletion of genes was due to a defect in cell growth or to their antiviral properties, an additional screen was performed in which Calu-3 cells presented with the CRISPR library were allowed to grow for 7 days, at the end of which those containing sgRNAs influencing viability were depleted (Supplementary Data 3)[57]. Comparison between sgRNAs depleted in these cells which were not subjected to infection to those depleted from the infected cells, identified several bonafide antiviral candidates such as MMP23B, LARS2, HUS1, STX4, and RER1 (Supplementary Data 2). Interestingly, the MMP23B metallopeptidase (MMPs) was reported to be downregulated in the lungs of COVID-19 patients. The reported downregulation of MMP23B in the presence of the virus may represent a host-pathogen crosstalk mechanism that includes a counter-reaction of the virus for avoiding cellular antiviral responses[58].

**Validation of the genetic screens**. To validate the candidate genes distinguished by the SARS-CoV-2 screens, individual Calu-3 KO cell lines were generated by expressing a subset of sgRNAs targeting specific genes which exhibited high levels of enrichment or depletion in the SARS-CoV-2 screens (enrichment score −log10 > 4) (Supplementary Data 1). Correct disruption of targeted genes was confirmed by DNA sequencing of their respective chromosomal loci or by RT-PCR analysis (Supplementary Table 1). The sensitivity of the mutated cells to SARS-CoV-2 infection was then addressed by quantification of the viral titer. Results show that disruption of many proviral candidates significantly impaired viral infection, while mutation of antiviral genes markedly increased SARS-CoV-2 RNA levels (Fig. 1e). In addition, ablation of proviral candidates protected cells from SARS-CoV-2-induced cell death while mutations in antiviral candidates resulted in increased cell death compared to infected WT cells (Fig. 1f).

Since several top proviral candidate genes identified in the SARS-CoV-2 screens (e.g., *GATA6, IRF6,* and *CUL5*) did not emerge as essential for SARS-CoV-2 infection in previously published CRISPR screens carried in other cell types, we addressed the possibility that the observed phenomenon is related to their expression in different cell types. Indeed, an inspection of the relative transcription levels of GATA6, IRF6, and CUL5 in published mRNA-seq datasets of Calu-3[59], A549[60], and Vero-E6[61] cell lines, revealed that GATA6 and IRF6 are expressed at much higher levels in Calu-3 cells (IRF6 expression is below threshold in Vero-E6 and A549) (Fig. 1g). CUL5 expression appeared to exhibit similar levels in various cell lines, therefore it is conceivable that its detection in the current screen, may be attributed to a Calu-3 cell-specific modulation in the course of infection. Taken together, the data therefore suggest that some of the host entry factors necessary for SARS-CoV-2 infection exhibit cell-line specificity, illustrating the importance of using relevant cell models to maximize the understanding of SARS-CoV-2 entry. Finally, an inspection of data from two recently documented Calu-3 infection screens[21,62] revealed that multiple top proviral candidates identified in our screens were enriched at a commensurate extent also in these independent screens (Fig. 1h).

Overall, corroboration of previously reported host-encoded viral-entry factors reported here, together with novel proviral host genes as well as the implied functional pathways defined by these factors, demonstrates the high potential of CRISPR-knockout screens to distinguish targets essential for viral pathogenicity, which may serve for designing host-dependent antiviral therapy.

**Functional pathways and interaction networks of SARS-CoV-2-host factors.** Gene set enrichment analysis (GSEA) confirmed that antiviral genes (depleted from the library) are strongly enriched for mitochondrial function, specifically mitochondrial translation and respiratory electron transport chain, and translation functions, including translation elongation and amino acid activation (Fig. 2a). Another large group of depleted genes was found to participate in the phosphatidylglycerol biosynthetic process. To examine whether these depleted host pathways affect the virus specifically or exert an indirect effect on cell viability, GSEA was performed also on the data of the control viability screen conducted with uninfected cells. The analysis revealed that most of the pathways suspected as antiviral are actually important for cell survival regardless of infection. The only pathway that was confirmed as specific antiviral was the phosphatidylglycerol biosynthetic process (normalized enrichment score (NES) = −1.94, FDR = 0.007 in the infection screen, NES = −1.35, FDR = 0.526 in the cell viability screen) (Fig. 2a). Interestingly, phosphatidylglycerol was reported to regulate the innate immune response in lungs[63] and consequently was suggested as a treatment against SARS-CoV-2 infection by restoring the lung tissue[64]. No functional pathways were found to be significantly represented by the analysis of enriched genes found as essential (proviral) for SARS-CoV-2 infection.

For further inspection of these proviral genes, STRING-db interaction network analysis was employed for the top 200 enriched genes (Fig. 2b). The connected network components annotated using REACTOME pathways functionally implicated several cellular pathways. A large group of these genes, including the largest cluster, is related to immune system functions, and includes the genes *NFKB1, IRF6, IL2Ra,* and *EP300*. In addition, the analysis identified gene groups belonging to lipid metabolism, and genes related to vesicle-mediated transport, both functions known to be important determinants of viral infection.

**Differential analysis of host-factors dependencies exhibited by variants of SARS-CoV-2.** Reported differences in the infectivity as well as in the progression of the disease caused by different variants of SARS-CoV-2 indicate that these emerging variants may require both common and distinct host factors. Identification of these shared and diverged host requirements may enable a better understanding of the host-pathogen interactions characterizing SARS-CoV-2 infection, and most importantly may facilitate the design of future therapies. Accordingly, a comparative analysis of the datasets of the screens involving either one of the SARS-CoV-2 variants: SARS-CoV-2 B.1.1.7 (Alpha), SARS-CoV-2 B.1.351 (Beta), and the original WT-SARS-CoV-2, was conducted. The individual variant analysis enabled the distinction of the top-scoring host genes involved in the infection of each SARS-CoV-2 lineage (Fig. 3a). The cellular entry factor, ACE2, was found to be highly enriched for all lineages, illustrating that the main mechanism for entry into the target cells is similar for the three SARS-CoV-2 variants and reiterating the robustness of the screens. Another host gene that was consistently highly enriched in the screens of cells infected by the three variants was the transcription factor GATA6. As of today, GATA6 role in SARS-CoV-2 infection was not reported and is extensively addressed below. Conversely, the analysis identified different genes specifically involved in the infection by either one of the variants. Such are *ARIH2* and *CHMP2B,* which exhibited a higher enrichment score in the screen of the cells infected by the Alpha and Beta variants, respectively. Additional significantly enriched genes are highlighted in Fig. 3a.

To further distinguish the role of these host genes in the infection of each of the variants, a pairwise correlation of the genome-wide datasets was conducted (Fig. 3b). In spite of the inherent variability of the experimental system, which may complicate the distinction of differential dependencies between screens in a comprehensive manner, the top hits, representing genes encoding for functions strongly affected by the infection, exhibited high reproducibility. This is demonstrated by the correlation analysis of gene enrichment between biological replicates (Supplementary Fig. 1). Thus, since only genes strongly enriched or depleted are considered, the present report provides only a partial landscape of the host factors involved in the different VOCs infection. The proviral genes, *ACE2,* and *GATA6* were consistently highly enriched in the screens of the cells infected with all variants, while the antiviral *MMP23B LARS2 HUS1 STX4,* and *RER1* were depleted for all three variants. Most interestingly, some genes were enriched in the screen conducted with one mutant while depleted in the screens carried out with other variants (Fig. 3b). Two such genes were *KREMEN2* and *SETDB1* that in the evaluation of the top five enriched genes of the Alpha variant screen (which exhibited the best correlation between replicates) revealed an opposite pattern in the screens involving the Beta variant and the WT-SARS-CoV-2, suggesting that the products of these genes fulfill a distinct role during the life cycle of the Alpha variant. (Fig. 3c). Notably, the proteins KREMEN1 and KREMEN2 have been identified as co-receptors for Dickkopf (Dkk) proteins, hallmark secreted antagonists of the canonical Wnt signaling pathway[65,66]. In addition, KREMEN1 serves as a host-cell entry receptor for a major group of Enteroviruses[67]. Remarkably, in a recent screen in search of additional SARS-CoV-2 receptors, KREMEN1 was identified as an alternative viral-entry receptor[68]. Thus, the observation documented here suggests that KREMEN2 may play a similar role specifically during the life cycle of the Alpha variant, consequently contributing to its enhanced infectivity compared to other SARS-CoV-2 variants.

SETDB1 (SET domain bifurcated histone lysine methyltransferase 1), is a histone H3K9 methyltransferase that contributes

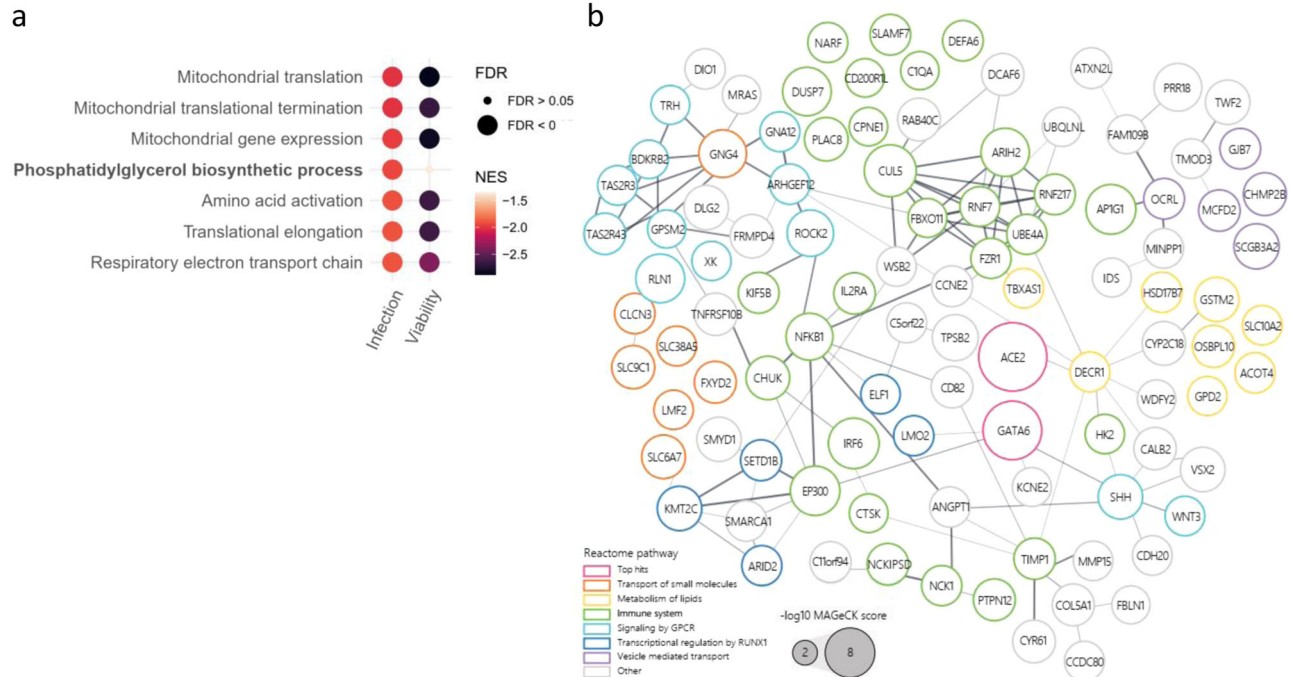

**Fig. 2 Gene set enrichment and interaction networks analysis identify functional pathways important for wild-type SARS-CoV-2 and variants of concern infection. a** Gene set enrichment analysis (GSEA) was performed by analysis of the directed scores of all genes from the WT-SARS-CoV-2 and VOCs screens combined (infection), and for the uninfected cell viability screen (viability). Results from the infection screens were filtered to get significantly enriched pathways with false discovery rate (FDR) of less than 0.01. The enrichment of each of these pathways in each screen is represented as bubbles colored by Normalized enrichment score (NES) and sized by significance levels. **b** Protein–protein interaction network for top 200 enriched hits from all strain screens combined based on STRING analysis. Edge thickness represents interaction score. Nodes are colored by selected REACTOME pathways and sized according to log10 transformed screen enrichment score. Source data are provided as a Source Data file.

together with TRIM28 to heterochromatin formation[69]. Recent studies have drawn attention to the involvement of SETDB1 in epigenetic control of the immune response, including antiviral response and IFN production[70–72]. Interestingly, the transcriptional levels of IFN-I, IFN-II, TRIM28, and SETDB1 were found to be elevated in SARS-CoV-2-infected children with mild symptoms, while their levels decreased in children with severe clinical pictures or MIS-C suggesting that they may play important roles in conditioning the evolution of the infection[73]. SETDB1 was also identified in a HCoV-NL63 genetic screen as a possible host chromatin regulator that promotes successful infection[22].

To validate KREMEN2 and SETDB1 specificity for the Alpha variant sgRNAs targeting either *KREMEN2* or *SETDB1* were expressed in Calu-3 cells that were subsequently infected by WT-SARS-CoV-2 or by either one of the VOCs. In both KREMEN2 and SETDB1 KO cell lines, a significantly impaired viral infection of the Alpha VOC was measured. However, disruption of either one of the genes did not result in a considerate change in viral titer by infection with the WT-SARS-CoV-2 or the Beta variant (Fig. 3d). While the data presented in this report clearly suggests that the *KREMEN2* and *SETDB1* genes encode functions specifically assisting the Alpha variant infection, additional studies will be required to understand their mechanism of function and strain specificity, and most importantly to determine if their role relates to possible enhanced viral infectivity or pathogenicity.

Overall, the comparative analysis shows that the survival-based CRISPR screen provides datasets of host factors common and distinct for a variety of SARS-CoV-2 lineages that should be further validated and taken under consideration for the design of broad-spectrum therapies for controlling SARS-CoV-2 infection.

**GATA6 is essential for SARS-CoV-2 infection**. The transcription factor GATA6, emerged as the second-strongest proviral host factor (inferior only to the viral receptor ACE2) for all three variants tested on the screens. GATA6 is a member of a small family of zinc-finger DNA-binding transcription factors that play an important role in the regulation of cellular differentiation. Of note, GATA6 was shown to promote the transcription of *SFTPA* gene, which is involved in immune and inflammatory responses, and lowers the surface tension in the alveoli[74,75]. Both *GATA6* and *SFTPA* genes were upregulated in SARS-CoV-2-infected lungs, while the GATA6 antagonist LMCD1 was downregulated[45]. To confirm the essentiality of GATA6 for SARS-CoV-2 infection, the expression of GATA6 in Calu-3 cells was abrogated by CRISPR-mediated targeting of GATA6, followed by inspecting the sensitivity of cells to SARS-CoV-2 infection. CRISPR-mediated ACE2 disruption in Calu-3 cells served as a positive control. The gene targeting manipulations indeed resulted in depletion of GATA6 and ACE2 expression as determined by real-time PCR (Supplementary Table 1). As expected, while WT Calu-3 cells display a cytopathic effect following WT-SARS-CoV-2 infection, ACE2 gene-disruption protected cells from the infection (Fig. 4a). In agreement with the results of the CRISPR screens, GATA6-disrupted cells were almost fully protected from cell death following WT-SARS-CoV-2 infection (Fig. 4a). Furthermore, the extent of resilience to viral infection was commensurate with that exhibited by cells in which ACE2 expression was abrogated (Fig. 4a). Further investigation of WT-SARS-CoV-2 viral load in infected GATA6-disrupted cells 24 and 48 h post-infection, demonstrated a significant reduction of viral replication subsequent to the reduction in GATA6 content (Fig. 4b). Further inspection of the effect of GATA6 disruption on the infection by the Alpha and Beta VOCs resulted in increased resilience to cell

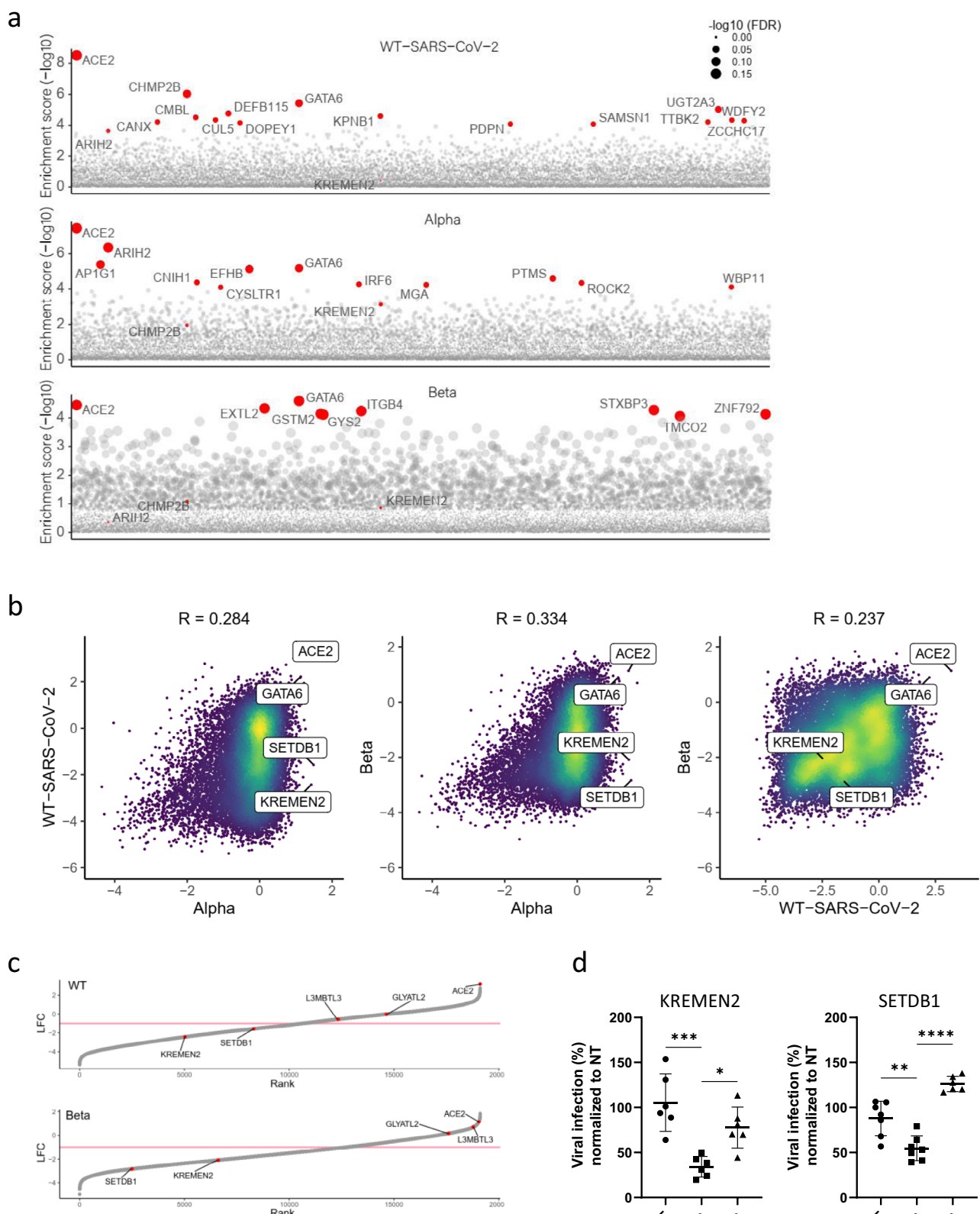

death of GATA6 KO cells as seen by microscopy imaging and crystal violet staining (Fig. 4c, d), and impaired viral infection (Fig. 4e). The effect of GATA6 gene disruption was also interrogated on the infectivity of the novel Delta (B.1.617.2) VOC (which emerged in the course of preparing the current report). The data in Fig. 4d, e establish that the Delta variant exhibited

impaired infectivity in the GATA6 KO cells confirming the strain-independent essentiality of GATA6 for SARS-CoV-2 infection.

As mentioned, previously published mRNA-seq transcriptomic datasets of Calu-3[59], A549[60], and Vero-E6[61] cell lines, revealed that GATA6 is expressed at much higher levels in Calu-3 cells (Fig. 1g), suggesting that GATA6 was not identified in screens

**Fig. 3 Differential analysis of variants of SARS-CoV-2 screens highlights shared and specific host-factors dependencies. a** Gene enrichment score for CRISPR screens of WT-SARS-CoV-2, Alpha, and Beta variants infection. Enrichment scores were determined by MaGECK. **b** Scatter plot depicting gene enrichment in WT-SARS-CoV-2, Alpha, and Beta variants screens. R, Pearson correlation. **c** Rank plot showing the log fold-change (LFC) of genes in the WT-SARS-CoV-2 screen (top) or the Beta variant screen (bottom) and its rank according to their LFC (lowest LFC ranked 1). Each gene is shown as a gray point. The top five most enriched genes in the Alpha variant screen (based on fold-change) are labeled and shown in red. A pink line marks LFC of −1. **d** WT-SARS-CoV-2, Alpha and Beta VOCs levels in infected KREMEN2 and SETDB1 knockout Calu-3 cells. Cells were infected using MOI = 0.002 for 48 h. A non-targeting (NT) sgRNA was used as control. Data were analyzed from six biologically independent experiments by one-way ANOVA with two-sided Dunnett's multiple comparisons test, comparing each experimental condition with Alpha variant. Shown are means ± SD. *$p = 0.011$; **$p = 0.0012$; ***$p = 0.0002$; ****$p < 0.0001$. Source data are provided as a Source Data file.

carried out in other cell lines due to cell-type specificity. To further substantiate this hypothesis, GATA6 was disrupted in Vero-E6 cells (Supplementary Fig. 2), and KO cells were subjected to infection of WT-SARS-CoV-2. Results demonstrated similar levels of infection between WT Vero-E6 cells and GATA6-disrupted cells (Fig. 4f). Cell viability following infection was also similar between the WT and GATA6 KO Vero-E6 cells (Fig. 4g). It is therefore conceivable that the function fulfilled by GATA6 in Calu-3 cells differs from that in Vero-E6, in line with the fact that GATA6 is a regulatory transcriptional factor that exerts its function by modulating the transcriptional level of various target genes, which may not be necessarily the same in different cell lines.

To further explore the physiologically and/or clinical relevance of GATA6 to SARS-CoV-2 infection, the level of GATA6 RNA in clinical nasopharyngeal-swab samples from 20 COVID-19 patients was analyzed and compared to that in samples collected from 20 healthy people. The analysis demonstrated that COVID-19 patients exhibited significantly higher levels of GATA6 than controls ($p < 0.05$) (Fig. 4h). Of note, the analysis did not reveal statistically relevant differences in ACE2 expression between COVID-19 and control healthy individuals (Fig. 4h). This observation further demonstrates that GATA6 may be involved in the susceptibility to viral infection, and suggests that GATA6 by itself is upregulated in response to infection by a regulatory inductive circuit different from that of ACE2 receptor. Upregulation of GATA6 was also observed in a recent study in which the host transcriptome from nasopharyngeal-swab samples of four COVID-19 patients was profiled[76]. In line with the above in vivo results, the relative levels of GATA6 transcription were found also to be elevated during SARA-CoV-2 infection of Calu-3 cells (Fig. 4i).

**GATA6 is a novel regulator of ACE2 that may serve as a potential host-directed therapeutic target.** To further elucidate which viral life-cycle step is facilitated by GATA6 expression, SARS-CoV-2 infection was visualized by immunofluorescent labeling of the virus in infected GATA6-disrupted cells in comparison to WT cells (Fig. 5a). The analysis enabled detection of SARS-CoV-2 in the cytoplasm of WT cells as soon as 5 h post-infection with increased viral load at 24 h post-infection. In contrast, a significant reduction of the infection of SARS-CoV-2 in the GATA6-disrupted cells was observed suggesting that GATA6 facilitates an early step in the viral life cycle possibly related to the entry of the virus into the cells (Fig. 5a).

Since SARS-CoV-2 entry into cells is primarily mediated by ACE2, the expression of ACE2 in ACE2 disrupted, WT and GATA6-disrupted cells was determined by western blot analysis. Notably, the decrease in GATA6 was followed by a significant reduction in ACE2 expression compared to control cells (Fig. 5b). Real-time PCR analysis confirmed the western blot results indicating that this effect involves a regulatory mechanism exerted at the transcription level, suggesting that GATA6 acts as a transcriptional factor affecting ACE2 expression (Fig. 5c). The data were validated by trans-complementation experiments,

which established that extrachromosomal cDNA-mediated expression of GATA6 (N-terminally triple FLAG-tagged and C-terminally triple AU1-tagged full-length human wt GATA6) in GATA6 KO cells restored elevated ACE2 expression, as measured by western blot analysis (Fig. 5b). Furthermore, complementation of GATA6 expression by exogenous GATA6 cDNA restored the susceptibility of the cells to SARS-CoV-2 infection as measured by the virus titers (Fig. 5d) and crystal violet staining (Fig. 5e). Based on the data it is possible therefore to assume that the low ACE2 levels associated with the abrogation of GATA6 expression are a manifestation of the role of GATA6 in viral entry. However, since GATA6 is a known pleiotropic transcription factor, its abrogation could theoretically affect the transcription of additional genes that may have a role in SARS-CoV-2 infection. In an attempt to further confirm that GATA6 is related to viral entry to the cell through regulation of ACE2 expression, GATA6 KO cells overexpressing ACE2 were infected with SARS-CoV-2. The data demonstrated that the expression of ACE2 in GATA6 KO cells restored SARS-CoV-2 titers suggesting that the main mechanism through which GATA6 influences SARS-CoV-2 infection is through modulation of ACE2 expression levels (Fig. 5d).

In an attempt to further determine the temporal connection between GATA6 expression and that of ACE2, the expression of GATA6 was abrogated by siRNA targeting, and the expression of ACE2 was interrogated at early time points post-GATA6 silencing. The analysis showed that ACE2 expression decreases as early as 12 h post-GATA6-siRNA transfection (Fig. 5f).

As a transcription factor GATA6 recognizes a well-defined DNA-binding specific consensus site (T/AGATAA/G or reverse complement sequences C/TTATCT/A)[77]. Accordingly, the promoter of the ACE2 gene was screened for the GATA6 consensus DNA-binding site. The conserved GATA-binding motif was found at nucleotide positions −341 and −403 (upstream of the initiation ATG codon) (Fig. 5g)[78]. To investigate the possible binding of GATA6 to ACE2 promoter an Electrophoretic Mobility Shift Assays (EMSA) was implemented. Incubation of lysates from HEK293T cells overexpressing GATA6 with labeled oligonucleotides derived from the promoter region of ACE2 resulted in the formation of electrophoretic slow-migrating protein-DNA complexes, which were not generated in the presence of competing unlabeled oligonucleotides (Fig. 5h). This observation supports the notion that GATA6 may induce the expression of ACE2 via direct binding at a sequence-specific GATA6 consensus site.

The connection between GATA6 and ACE2 expression, together with the observations that GATA6 transcription levels were upregulated in SARS-CoV-2-infected cells (Fig. 4i) and SARS-CoV-2-positive clinical samples (Fig. 4h) raised the possibility that overexpression of GATA6 might increase cell susceptibility to viral infection. To address this possibility, Calu-3 cells overexpressing GATA6 were subjected to SARS-CoV-2 infection. Western blot analysis demonstrated that the elevated levels of GATA6 expression were accompanied by increased levels of ACE2 (Supplementary Fig. 3a). Yet, the overexpression of

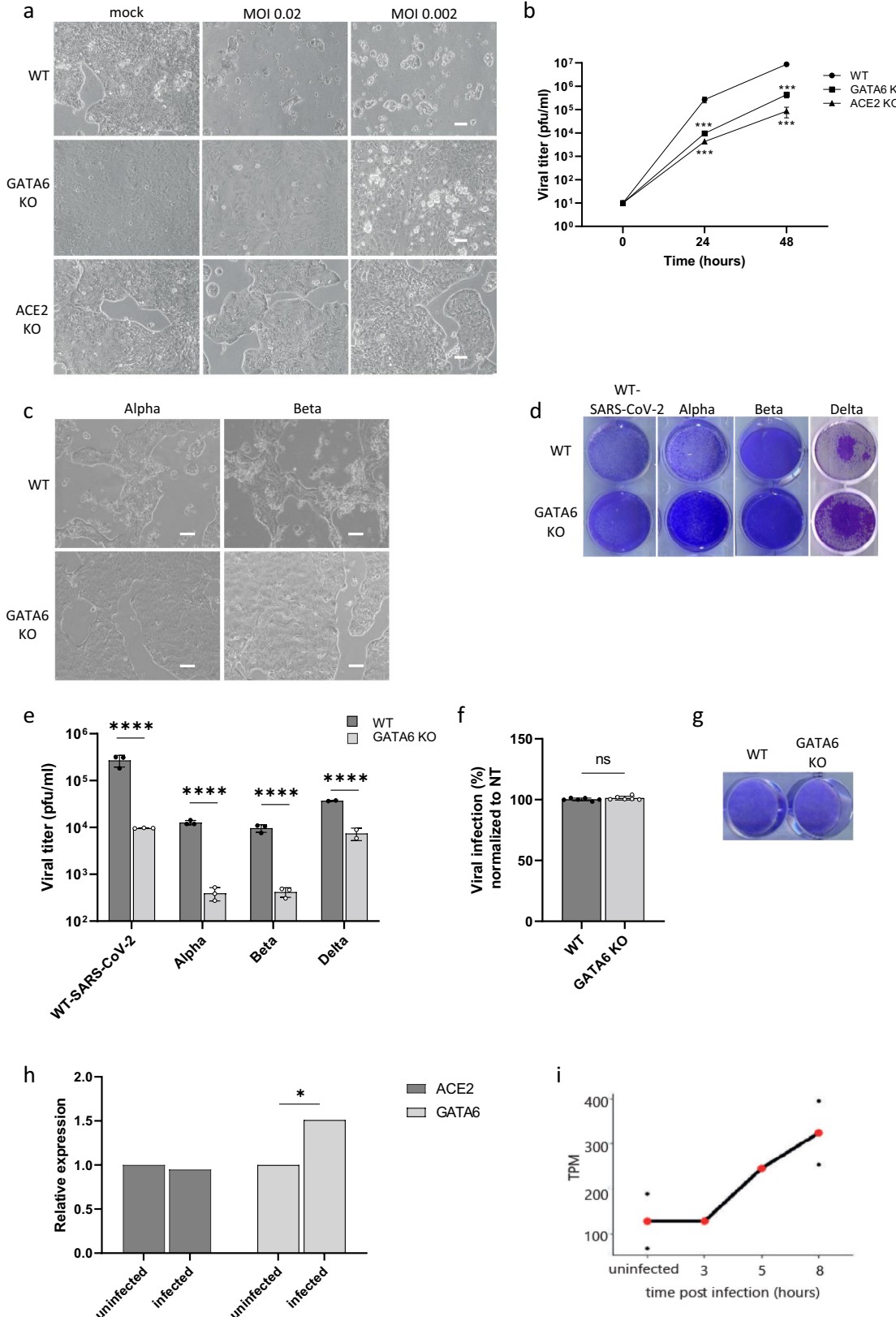

GATA6 did not affect the RNA levels of SARS-CoV-2 measured 48 h post-infection (Supplementary Fig. 3b).

CRISPR loss-of-function surveys for host factors based on virus-induced cell death, may identify genes encoding for products essential to the virus, and yet dispensable to cell viability. Accordingly, such host factors exhibiting specific roles in

the viral pathogenicity may serve as appropriate candidates for antiviral therapy with minimal host side effects. Therefore, we explored whether pharmacological inhibition of GATA6 affects ACE2 expression and consequently, coronavirus infection. Pyrrothiogatain, previously reported to inhibit the DNA-binding activity of GATA6 and other members of the GATA

**Fig. 4 GATA6 is a proviral host factor that is upregulated in response to infection. a** Control, GATA6, and ACE2 disrupted Calu-3 cells were infected with WT-SARS-CoV-2 at the indicated MOI. Cell viability was visualized by light microscopy 48 h post-infection. Representative images of three independent replicates are shown. **b** Supernatants were collected from WT, ACE2, and GATA6 knockout cells infected with SARS-CoV-2 at the indicated time points. Virus production, as measured by plaque-forming units (PFU) per milliliter, was determined by plaque assay. Data were analyzed from three biologically independent experiments by two-way ANOVA with two-sided Dunnett's multiple comparisons test, comparing each KO cells sample to the WT. Shown are means ± SD. *$p < 0.0001$. **c** Control and GATA6-disrupted Calu-3 cells were infected with the Alpha or the Beta VOCs at MOI = 0.002. Cell viability was visualized by light microscopy 48 h post-infection. Representative images of three independent replicates are shown. **d** Cell viability assay of control and GATA6-disrupted Calu-3 cells infected with the indicated SARS-CoV-2 variant at MOI of 0.002 for 48 h, and stained with crystal violet. One of three repetitions is shown. **e** Real-time PCR quantification of WT-SARS-CoV-2, Alpha, Beta, and Delta VOCs levels in infected control and GATA6-disrupted Calu-3 cells. Cells were infected using MOI = 0.002 for 48 h. Data were analyzed from three biologically independent experiments by one-way ANOVA with two-sided Šídák's multiple comparisons test. Shown are means ± SD. ****$p < 0.0001$. **f** real-time PCR quantification of control and GATA6-disrupted Vero-E6 cells infected with WT-SARS-CoV-2 at MOI of 0.002 for 48 h. Shown are means ± SD. Data were analyzed from six biologically independent experiments by two-tailed student's $t$-test. $p = 0.19$. ns not significant. **g** Cell viability assay of control and GATA6-disrupted Vero-E6 cells infected with WT-SARS-CoV-2 at MOI of 0.002 for 48 h, and stained with crystal violet. One of three repetitions is shown. **h** real-time PCR measurements of ACE2 and GATA6 mRNA levels relative to GAPDH amount in negative (uninfected) and positive (infected) nasopharyngeal-swab samples from symptomatic and asymptomatic individuals Data were analyzed by two-tailed student's $t$-test, *$p < 0.05$. **i** Expression levels of GATA6 at different times along SARS-CoV-2 infection in Calu-3 cells[67] normalized to transcripts per million (TPM) values. Values from single replicates shown for 3 h and 5 h. Mean values of duplicates shown for uninfected and 8 h in red, and individual replicate values are presented as black points. Source data are provided as a Source Data file.

family[79], was tested for its ability to induce downregulation of ACE2 and subsequent inhibition of viral infection.

As expected, the drug exhibited a dose-dependent inhibition of GATA6 binding to ACE2 as shown by Electrophoretic Mobility Shift Assays (Fig. 6a, note also the increase in the amount of the free unbound DNA probe upon competition with unlabeled sequences, which confirms the sequence specificity of the binding). Notably, Calu-3 cells retained their viability even at high doses of Pyrrothiogatain (Fig. 6b). Inhibition of GATA6 resulted in lower ACE2 expression (Fig. 6c) supporting the causal connection between GATA6 and ACE2. A decrease in the amount of GATA6 protein was also observed (Fig. 6c) suggesting that Pyrrothiogatain may induce a feedback circuit lowering GATA6 and/or a conformational change affecting its expression. Most importantly, treatment of infected Calu-3 cells with Pyrrothiogatain resulted in a significant decrease in the viral load of the culture (Fig. 6d). Immunofluorescence analysis further evidenced cytoplasmic SARS-CoV-2 staining in untreated cells while the significantly reduced infection was observed in Pyrrothiogatain-treated cells (Fig. 6e). The data cannot rule out the possibility that other members of the GATA family are affected by Pyrrothiogatain, yet other previous transcriptomic surveys neither evidence the upregulation of other GATAs nor did our screen distinguish the enrichment of other GATAs upon infection (Supplementary Data 1). Furthermore, with the exception of GATA6, none of the other 5 members of the GATA family revealed significant levels of transcription in the Vero-E6, A549, and Calu-3 cell lines (Fig. 6f).

The data strongly support the possibility that novel therapeutic strategies to countermeasure viral infection via down-modulation of ACE2 may employ specific GATA6 targeting. GATA6 targeting may seem a rather unlikely therapeutic approach considering the fact that it is a pleiotropic transcription factor affecting many loci. Yet, recently, the involvement of transcription factors in many pathologies prompted a preclinical and clinical assessment of their therapeutic value, including studies addressing proteins belonging to the GATA family[80–82]. The possibility to target transcription factors for selective therapeutic intervention, is facilitated by high combinatorial interplay and compensations of the transcription factors which lowers the side effects associated with targeting one particular factor.

Since transcription factors (including GATA6) may play different roles in different cell types (for example, GATA6 acts as a tumor suppressor in astrocytoma while it is overexpressed in

human colon cancer and pancreatic carcinoma), some of these studies, including those focusing on GATA family members, mitigate the undesired effects by combining specificities in gene and tissue targeting.

In summary, the screens conducted in human lung epithelial cells, provided a comprehensive catalog of cellular factors and functional pathways critical for the infection of WT-SARS-CoV-2 and additional VOCs. These include known and novel host factors such as the viral receptor ACE2 and various components belonging to the Clathrin-dependent transport pathway, ubiquitination, and Heparan sulfate biogenesis. In addition, the comparative analysis highlights commonalities and differences between SARS-CoV-2 variants and enables the identification of the receptor KREMEN2 and SETDB1 as possible unique genes required only for the Alpha variant. The differences exhibited by the viral variants with respect to the essentiality of specific host factors for their life cycle is of outmost importance since they may provide an explanation for the reported differences in their pathogenicity, infectivity, and disease progression.

Finally, the study evidenced the requirement of GATA6 for infection by WT-SARS-CoV-2 as well as the Alpha and Beta variants. Furthermore, analysis of clinical samples of COVID-19 infected patients showed an elevated level of GATA6, suggesting that in the course of the disease, a viral-induced modulation of the level of GATA6 occurs. Further investigations revealed that GATA6 regulates ACE2 expression and is critical for the entry step of SARS-CoV-2. Notably, pharmacological inhibition of GATA6 results in down-modulation of ACE2 and consequently inhibition of the viral infectivity suggesting that this protein may serve as a therapy target. These observations together with the loss-of-function screen data reported here contribute to a better understanding of SARS-CoV-2 pathogenesis and may represent an important basis for the future development of host-directed therapies.

## Methods

**Cell lines.** HEK293T (ATCC- CRL-3216), Vero-E6 (ATCC- CRL-1586), and Calu-3 cells (ATCC- HTB-55) were cultured in Dulbecco's Modified Eagle Medium (DMEM) with 10% heat-inactivated fetal bovine serum (FBS), 10 mM non-essential amino acids (NEAA), 2 mM L-Glutamine, 1 mM sodium pyruvate and 1% Penicillin/Streptomycin (all from Biological Industries, Israel) at 37 °C, 5% $CO_2$. All cell lines tested negative for mycoplasma.

**Virus stocks.** The original SARS-CoV-2 (GISAID accession EPI_ISL_406862) was kindly provided by Bundeswehr Institute of Microbiology, Munich, Germany.

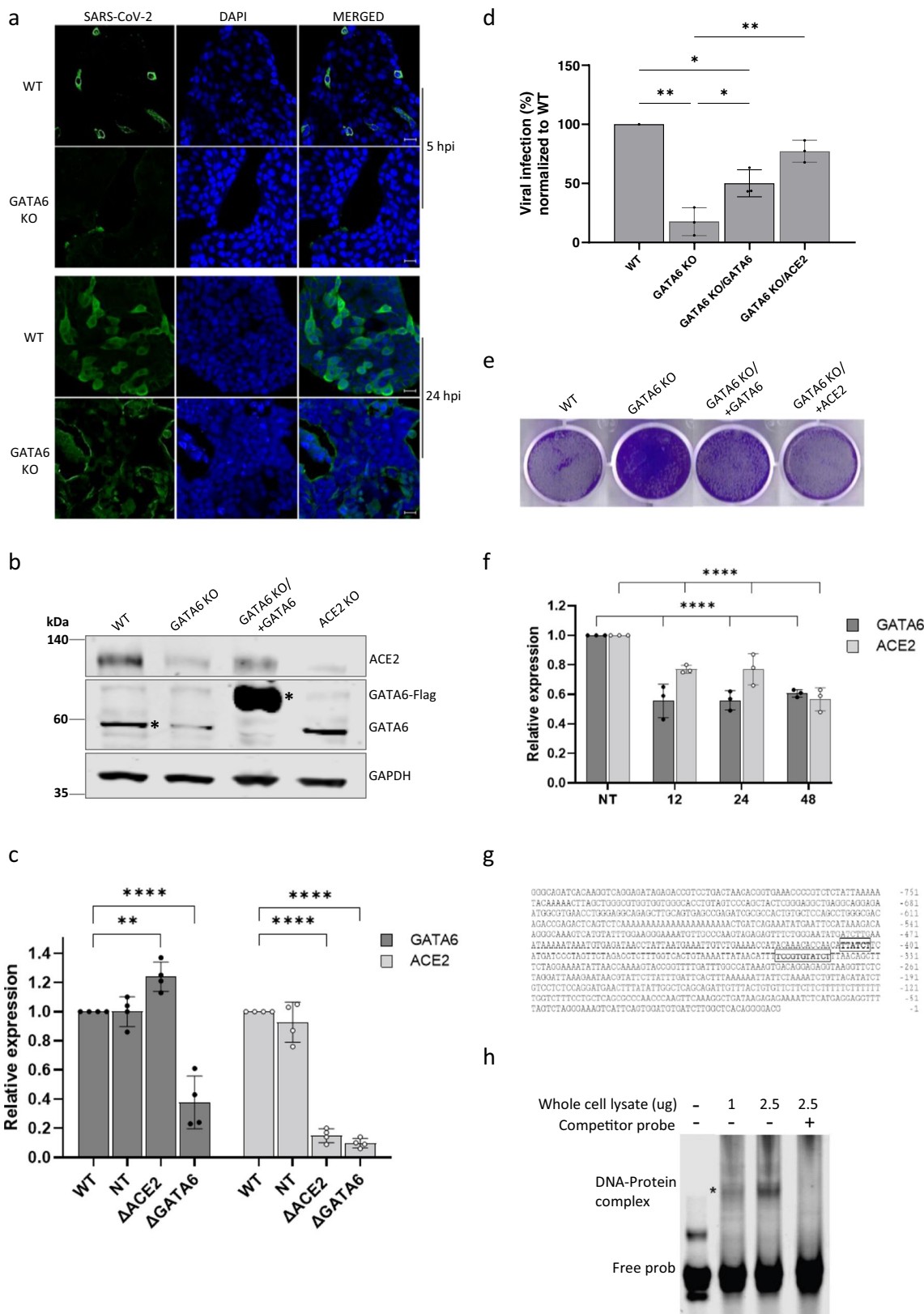

SARS-CoV-2 B.1.1.7 (Alpha) and B.1.351 (Beta) and B.1.617.2 (Delta) variants of concern (VOCs) were kindly provided by Michal Mandelboim, MOH Central Virology Lab, Tel Hashomer, Israel. Original, B.1.351 and B.1.617.2 VOC stocks were propagated (2–3 passages) on Vero-E6 cells while B.1.1.7 was propagated on Calu-3 cells. All viruses were tittered on Vero-E6 cells. Handling and working with SARS-CoV-2 virus were conducted in a BSL3 facility in accordance with the bio-safety guidelines of the Israel Institute for Biological Research (IIBR).

**Genome-wide CRISPR screens**. To generate CRISPR KO libraries, a total of $4 \times 10^8$ Calu-3 cells were transduced with lentivirus of human Brunello Human CRISPR library (Addgene #73179, gift from David Root and John Doench) in the presence of 0.5 mg/ml polybrene (TR-1003, Sigma), at a MOI of 0.3. Two days post-transduction, 5 ug/ml puromycin (ant-pr-1, InvivoGen) was added to the media, and transduced cells were selected for seven days. Twenty-four hours prior to infection with SARS-CoV-2, $1.5 \times 10^7$ Calu-3 library-cells were seeded in 75 cm²

**Fig. 5 GATA6 is a novel regulator of ACE2. a** WT-SARS-CoV-2-infected control and GATA6-disrupted Calu-3 cells were fixed 5 and 24 h post-infection and stained with antisera against SARS-CoV-2 (green) and DAPI (blue). Scale bar, 20 mm. Representative images of three biologically independent experiments are shown. **b** Western blot for the detection of GATA6 and ACE2 levels was performed in WT, ACE2 knockout, GATA6 knockout, and complemented Calu-3 cells. Representative images of two biologically independent experiments are shown. **c** ACE2 and GATA6 mRNA levels relative to GAPDH amount were analyzed in WT, NT control, ACE2 and GATA6 knockout Calu-3 cells by real-time PCR. Data were analyzed from four biologically independent experiments by one-way ANOVA with two-sided Tukey's multiple comparison test. Shown are means ± SD. **$p < 0.001$; ****$p < 0.0001$. **d** Real-time PCR quantification of WT-SARS-CoV-2 in infected WT, GATA6 knockout and GATA6 knockout complemented by GATA6 or ACE2 expression Calu-3 cells. Cells were infected using MOI = 0.002 for 48 h. Data were analyzed from three biologically independent experiments by one-way ANOVA with two-sided Tukey's multiple comparison test. Shown are means ± SD. *$p < 0.05$; **$p < 0.005$. **e** Cell viability of control, GATA6 knockout and GATA6 knockout complemented by GATA6 or ACE2 cells infected with WT-SARS-CoV-2 at MOI of 0.002 for 48 h, and stained with crystal violet. One of three repetitions is shown. **f** Cells transfected with a control or siRNAs targeting GATA6 were analyzed by real-time PCR for GATA6 and ACE2 expression relative to GAPDH amount. Data were analyzed from three biologically independent experiments by one-way ANOVA with two-sided Tukey's multiple comparison test. Shown are means ± SD. ****$p < 0.0001$. **g** ACE2 ATG codon upstream sequences from nucleotide −1 to −820 are shown. Sequences shown framed in bold are potential GATA-binding sites. **h** Electrophoretic Mobility Shift Assays (EMSA) of lysates from HEK293T cells overexpressing GATA6 with labeled oligonucleotides derived from the promoter region of ACE2 results in the formation of protein-DNA complexes, which was prevented by competitive unlabeled oligonucleotides. Representative images of two biologically independent experiments are shown. Source data are provided as a Source Data file.

flasks. For a control reference for sgRNA enrichment analysis cells were harvested 48 h after seeding. For viability screen, mock cells were harvested 7 days after seeding and served as a control for the depletion of genes that are essential for cell survival. Three screens, using three variants of SARS-CoV-2, were performed with $3 \times 10^7$ cells in each screen which is sufficient for the representation of each sgRNA into ~400 unique cells. All screens were performed in duplicate. For SARS-CoV-2 infection cells were washed once with RPMI without FBS and infected with SARS-CoV-2 virus, at a MOI of 0.02–0.04, in the presence of 20 μg per ml TPCK trypsin (Thermo scientific) and 2% FBS. Flasks were incubated for 1 h at 37 °C to allow viral adsorption. Then, RPMI medium supplemented with 2% FBS, was added to each well[83]. Nine days post-infection with SARS-CoV-2, surviving cells pellet was dissolved in 10% triton and boiled for 30 min. Genomic DNA (gDNA) was extracted using a QIAamp DNA mini kit (Qiagen). sgRNA sequences were amplified by one-step PCR using primers with illumine adapters and the gDNA as a template. A master mix consisted of 75 μl ExTaq DNA Polymerase (Clontech), 1000 μl of 10× ExTaq buffer, 800 μl of dNTP provided with the enzyme, 50 μl of P5 stagger primer mix (stock at 100 μM concentration), and 2075 μl water. Each PCR reaction consisted of 50 μl gDNA, 40 μl PCR master mix, and 10 μl of uniquely barcoded P7 primer (stock at 5 μM concentration). PCR cycling conditions: an initial 1 min at 95 °C; followed by 30 sec at 95 °C, 30 sec at 53 °C, 30 sec at 72 °C, for 28 cycles; and a final 10 min extension at 72 °C; hold at 4 °C. The p5 stagger primer and uniquely barcoded P7 primers were synthesized at Integrated DNA Technologies, Inc. Primers sequences are listed in Supplementary Table 2. Pooled PCR products were purified with Agencourt AMPure XP-SPRI magnetic beads according to the manufacturer's instruction (Beckman Coulter, A63880). Samples were sequenced on a Illumina MiSeq platform. Reads were counted by alignment to a reference file of all possible sgRNA present in the library. The read was then assigned to a condition on the basis of the 8 nt index included in the p7 primer. The lentiviral plasmid DNA pool was sequenced as a reference.

**Analysis of CRISPR-Cas9 genetic screen data**. MAGeCK v0.5.6[84] was used to count sgRNA from FASTQ files and to analyze the selection effect of genes based on the change in sgRNA distribution, using the robust rank aggregation (RRA) algorithm with normalization to total reads. The directed score for each gene was calculated by taking the enrichment MAGeCK RRA scores for genes with a positive fold-change from control and the depletion MAGeCK RRA score for genes with a negative or zero-fold change. Directed scores for sgRNAs were calculated by multiplying the MAGeCK sgRNA scores by 1 or (−1) according to the direction of change. These scores were scaled to sd = 1 and centered on the mean. All genes and their enrichment scores can be found in Supplementary Data 1.

**Pathway and network analysis**. Directed scores calculated from MAGeCK RRA scores of all screens combined (treating all screens as one experiment) were used as input for gene set enrichment analysis (GSEA version 4.1) with GO biological process (c5.bp) from MSigDB (version 7.4)[85,86]. Results were filtered to get significantly enriched pathways with a false discovery rate of less than 0.01. To further analyze functional pathways enriched in positive screen hits, the 200 genes that ranked the highest based on MAGeCK enrichment score were selected and used as input for STRING protein–protein interaction network analysis[87] using default parameters. The STRING network was imported into Cytoscape[88]. Genes were highlighted based on association with selected REACTOME pathways. Genes that were not highlighted or connected to at least two other genes were excluded from the graph.

**Generation of Vero-E6 and Calu-3 KO cell lines**. DNA oligos (Integrated DNA Technologies, Inc.) containing sgRNA sequences (see Supplementary Table 2) were

annealed and ligated into lentiCRISPRv2 (Addgene, #52961, gift from Feng Zhang). Lentivirus was packaged by co-transfection of constructs with the 2nd generation packaging plasmids pMD2.G and PsPax using jetPEI (Polyplus-transfection) into six-well plates with HEK293T cells according to protocol. Sixty hours post-transfection supernatants were collected, centrifuged at 1500 rpm for 5 min, and filtered through a 0.45 μm filter. Calu-3 or Vero-E6 cells were transduced with lentiviruses in the presence of 7 ug/ml polybrene (TR-1003, Sigma) and then selected with puromycin for 7 days. Knockout was confirmed by western blot, Real-Time PCR, or sequencing. For GATA6 complementation experiment, GATA6 knockout cells were complemented by transfection of 2.5 μg pBabe 3XFLAG-wt GATA6-3XAU1 puro vector (Addgene #72607) using Lipofectamine 3000 (L3000015 ThermoFisher) according to manufacturer's instructions. Seventy-two hours post-transfection, the expression of GATA6 was analyzed by western blot.

**Infection of Vero-E6 and Calu-3 KO cell lines**. Cells were seeded 48 h prior to infection at a density of $4 \times 10^5$ in 12-well plates and infected as described above at a MOI of 0.002. Plates were incubated for 1 h at 37 °C to allow viral adsorption. The medium was then aspirated and replaced with fresh medium. At that point (t = 0 h), 24 hpi and 48 hpi cell medium was taken for viral titration.

**Viral titration**. Vero-E6 cells were seeded in 12-well plates ($5 \times 10^5$ cells/well) and grown overnight in a growth medium. Serial dilutions of SARS-CoV-2 were prepared in an infection medium (MEM containing 2% FBS with NEAA, glutamine, and Penicillin/Streptomycin), and used to infect Vero-E6 monolayers in duplicates or triplicates (200 μl/well). Plates were incubated for 1 h at 37 °C to allow viral adsorption. Then, 2 ml/well of overlay [MEM containing 2% FBS and 0.4% tragacanth (Merck, Israel)] was added to each well, and plates were incubated at 37 °C, 5% $CO_2$ for 72 h. The media were then aspirated, and the cells were fixed and stained with 1 ml/well of crystal violet solution (Biological Industries, Israel). The number of plaques in each well was determined, and SARS-CoV-2 titer was calculated. For real-time PCR quantification of the virus a PFU calibration curve that was tested in parallel was utilized to express Ct values as calculated PFUs.

**Imaging and Immunofluorescence assay**. Calu-3 knockout or WT cells were seeded at a density of $4 \times 10^5$ in 12-well plates, infected as described above, or left uninfected. Imaging was performed on a Nikon eclipse TS100 microscope using a WD 7.0 10X/0.25 Ph1 DL objective and DS-Ri1 camera. For Immunofluorescence experiments, Calu-3 knockout or WT cells were seeded at a density of $5 \times 10^4$ cells per well in eight-well-μ-slides (Ibidi), infected as described above or left uninfected. Cells were washed once with PBS, fixed with 4% paraformaldehyde (PFA) in PBS for 15 min, and permeabilized using 0.5% Triton X-100 (Sigma T9284) for 2 min. The fixed cells were blocked in PBS containing 2% FCS and stained with primary antibodies (diluted 1:200 in blocking buffer) for 1 h at room temperature. After washing with PBS, cells were incubated with Alexa Fluor 488 conjugated anti-rabbit for 0.5 h. Nuclei were stained with DAPI (200 μg/ml D9542 Sigma) for 5 min at room temperature. Imaging was performed on an LSM 710 confocal scanning microscope (Zeiss, Jena, Germany). Primary antibodies used in this study: Rabbit anti-GATA6 (Abcam ab175349), hyperimmune Rabbit serum from intervenous (i.v) SARS-CoV-2-infected Rabbits.

**Cell viability assays**. The effect of Pyrrothiogatain on Calu-3 cell proliferation was determined using the XTT assay (Cell Proliferation Kit, 20-300-1000, Biological Industries, Bet Haemek, Israel) as per the manufacturer's instructions. Briefly, $10^5$ Calu-3 WT cells were seeded in 96-well plate. Twenty-four hours later, the cells were incubated with Pyrrothiogatain at different concentrations (150, 300, and

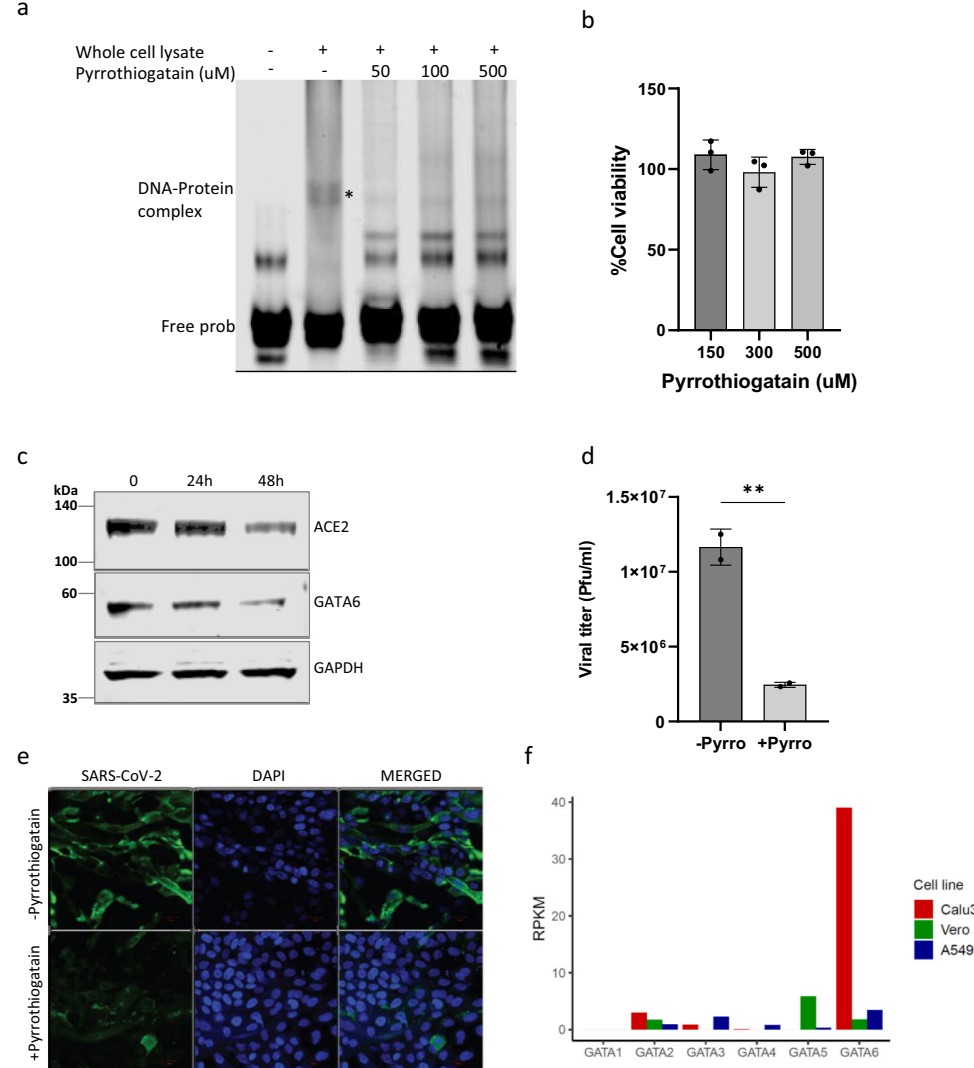

**Fig. 6 GATA6 may serve as a potential host-directed therapeutic target. a** Electrophoretic mobility shift assay (EMSA) of lysates from HEK293T cells overexpressing GATA6 with labeled oligonucleotides derived from the promoter region of ACE2 in the presence of various concentrations of Pyrrothiogatain (0–500 μM). **b** The effect of various concentrations of Pyrrothiogatain on Calu-3 cell proliferation was determined using the XTT assay and presented as a percentage of cell viability of untreated cells. Data were analyzed from three biologically independent experiments by one-way ANOVA with Brown-Forsythe test. **c** Calu-3 cells were treated with 500 uM Pyrrothiogatain and the expression of GATA6 and ACE2 were analyzed by western blot at the indicated time. **d** Real-time PCR quantification of WT-SARS-CoV-2 in Calu-3 cells that were pretreated with 500 uM Pyrrothiogatain. Cells were infected using MOI = 0.002 for 48 h. Data were analyzed from two biologically independent experiments by two-tailed student's $t$-test. Shown are means ± SD. **$p < 0.01$. **e** Calu-3 cells pretreated with 500 uM Pyrrothiogatain and infected with WT-SARS-CoV-2. Cells were fixed 24 h post-infection and stained with antisera against SARS-CoV-2 (green) and DAPI (blue). Scale bar, 20 mm. **f** mRNA read densities (reads per kilobase million, RPKM) of the six members of the GATA family in uninfected Calu-3 (red), Vero-E6 (green), and A549 cells (blue)[66–68]. Representative images of three biologically independent experiments are shown. Source data are provided as a Source Data file.

500 μM) for an additional 48 h. Then, XTT reagent was added for 15 min before reading the change in absorbance at 650 and 475 nm using the SpectraMax 250 plate reader (Molecular devices). Specific absorbance measurements were given as the mean ± SD absorbance calculated from two repeat wells/samples. The mean specific absorbance was normalized at each time point to that of the non-treated control.

For crystal violet staining, following 48 h of SARS-CoV-2 infection the medium was removed from the cells that were then fixed and stained with 0.1% crystal violet solution (Biological Industries, Israel) for 5 min. Then stain was aspirated and plates were rinsed once with tap water and air-dried.

**Analysis of GATA6, CUL5, and IRF6 expression in different cell lines**. Published mRNA-seq data used for A549, Vero-E6, and Calu-3 cells are from Kinori et al. 2016, Finkel et al. 2021a and Finkel et al. 2021b., respectively. Gene level values were normalized to TPM (transcripts per million) by dividing the published RPKM values by the sum of each sample and multiplying by one million.

**Analysis of clinical samples**. The research complies with all relevant ethical regulations and was approved by the Israel Institute for Biological Research (IIBR) IBC ethical committee according to the Israel Ministry of Health Covid-19 studies guideline. Ethical review and approval were waived since the samples used for this study were leftovers of anonymized samples. Negative and positive qRT-PCR nasopharyngeal-swab samples from symptomatic and asymptomatic individuals were collected as part of routine scanning of nursing homes. No information is available on the level of symptoms manifested by each tested positive individual. Viral RNA was extracted using RNAdvance Viral XP kit (Beckman Coulter). From each sample 200 μL were added to LBF lysis buffer, and further processed on the Biomek i7 Automated Workstation (Beckman Coulter), according to the manufacturer's protocol. Each sample was eluted in 50 μL of RNase-free water. Real-time RT-PCR assays were performed using the SensiFAST™ Probe Lo-ROX one-step kit (Bioline). In each reaction the primers final concentration was 600 nM and the probe concentration was 300 nM. Thermal cycling was performed at 48 °C for 20 min for reverse transcription, followed by 95 °C for 2 min, and then 45 cycles of 94 °C for 15 sec, 60 °C for 35 sec. Primers and probes (listed in Supplementary

Table 2) were designed using the Primer Express Software (Applied Biosystems) and purchased from Integrated DNA Technologies, Inc.

**siRNA inhibition assay**. Calu-3 Cells were transfected with siRNA validated for GATA6 or negative control (TriFECTa Kit DsiRNA Duplex, IDT) (listed in Supplementary Table 2) in the presence of Lipofectamine RNAiMAX reagent (Life Technologies), according to the manufacturer's standard protocol. At the indicated time after transfection, cells were harvested. All experiments were performed in triplicate, and representative results are reported.

**RT-PCR**. Total RNA was extracted using RNeasy Mini Kit (Qiagen 74104), reverse transcribed using qScript cDNA Synthesis kit (95047-025, Quanta bio) according to protocols, and subjected to real-time PCR analysis using perfecta SYBR Green FastMix Low ROX (Quanta bio 95074-250). Data shown are the relative abundance of the indicated mRNA normalized to that of GAPDH. Gene-specific primers are listed in Supplementary Table 2.

**Western blot**. Cells were lysed in RIPA Lysis Buffer (Merck 20-188) in the presence of cOmplete protease inhibitor cocktail (Roche 11697498001). Lysates were nutated at 4 °C for 10 min, then centrifuged at $20,000 \times g$ for 15 min at 4 °C. Equal amounts of cell lysates were denatured in 4× Laemmli sample buffer (Bio-RAD #1610747), separated on 4–12% NuPAGE Bis-Tris gels (invitrogen), blotted onto nitrocellulose membranes and immunoblotted with primary antibodies αGATA6 (Abcam ab175349, 1:500, ab22600, 1:500), αGAPDH (Cell Signaling 14C10, 1:2000), αACE2 (Sino Biological #10108-T60, 1:1000). The secondary antibody used was IRDye® 800CW conjugated Goat anti-Rabbit (Licor, 1:20,000). Reactive bands were detected by Odyssey CLx infrared imaging system (Licor). Protein concentration was measured by the BCA protein assay kit (Pierce 23225). Protein quantification was performed on Licor software. Uncropped and unprocessed scans are presented in the source data file.

**EMSA**. Electrophoretic mobility assay (EMSA). HEK293T cells were transfected with 3 μg pBabe 3XFLAG-wt-GATA6-3AU1 puro plasmid (Addgene #76207) using Lipofectamine 3000 (L3000015 ThermoFisher) according to the manufacturer's instructions. Forty-eight hours post-transfection, cells were lysed in RIPA Lysis Buffer (Merck 20-188) in the presence of cOmplete protease inhibitor cocktail (Roche 11697498001) and nutated at 4 °C for 10 min. The cell lysate was cleared by centrifugation, concentrated, and dialyzed into PBS using Amicon Ultra-4 30 K (UFC803024, Merck). The EMSA was preformed using two sets of double-stranded DNA oligonucleotides of ~40 bp spanning the putative GATA6 binding sites in the promotor region of ACE2. 5′-IRDye700-labeled single-stranded nucleotide probes or PCR primers, as detailed in supplementary Table 2, were purchased from Integrated DNA Technologies, Inc. Complementary single-stranded nucleotides were mixed at 100 nM concentration, heated at 100 °C for 5 min, and then cooled down to room temperature to allow duplex formation. Binding reactions were carried out using the Odyssey® EMSA Buffer Kit (829-07910, Li-Cor) according to the manufacturer's instructions, in the presence of NP-40 and $MgCl_2$. In competition assays, unlabeled competitor probes were included in the binding reactions in 200-fold excess relative to labeled probes. For inhibition of the binding of GATA6 to the promotor region of ACE2 Pyrrothiogatain was included in the binding reaction in 50,100 or 500 uM. The reactions were incubated for 30 min at room temperature in the dark. 1× Orange dye was added, and the samples were run on 8% TBE gel in 0.5% TBE buffer (LC6775, Novex) at 70 V for 90 min in the dark and imaged on an Odyssey® infrared imaging system (Li-Cor Biosiences).

**GATA6 inhibition assay**. Pyrrothiogatain (3-(2,5-dimethyl-1H-pyrrol-1yl) thiophene-2- carboxylic acid) was purchased from Santa Cruz (cat #sc-352288A). Cells were treated with the indicated concentration of Pyrrothiogatain (50 mM stock dissolved in DMSO) for 48 h prior to infection with SARS-CoV-2. Infection and subsequent immunofluorescence imaging were carried out as described above.

**Statistics and reproducibility**. Data were analyzed in Prism 9.2 (GraphPad). Unless otherwise stated, all representative data came from one of at least two independent experiments. Mean values are shown with SD error bars. The relevant statistic test and $P$-values are described for each figure in the legends. No data were excluded from the analyses.

**Reporting summary**. Further information on research design is available in the Nature Research Reporting Summary linked to this article.

## Data availability

All data generated or analyzed during this study are included in this published article and its supplementary information files. The sgRNA count raw data generated in this study have been deposited in Gene Expression Omnibus under accession code GSE197962. The processed screen results generated in this study are provided in Supplementary Data 1-3.

Calu-3 expression data analyzed in this study are available from Gene Expression Omnibus with the accession code GSE162323. Vero-E6 expression data analyzed in this study are available from Gene Expression Omnibus with the accession code GSE149973. A549 expression data analyzed in this study are available from Gene Expression Omnibus with the accession code GSE82232. Source data are provided with this paper.

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

## Acknowledgements

We thank Emanuelle Mamroud for providing valuable feedback; Tamar Aminov and Inbar Chomsky for technical assistance; and Shay Weiss for biosafety guidance.

## Author contributions

M.I., Y.F., N.P., O.C., N.S.-G., and A.B.-K. conceptualized the study; M.I., O.I., M.A., R.F., U.E., and A.B.-K. performed experiments; I.N., L.K., and M.M. provided reagents. Y.F., I.C.-G., A.B.-D., S.R., and A.B.-K. undertook data analysis; Y.Y.-R., N.P., and T.I. performed experiments with SARS-CoV-2; M.I., Y.F., T.C., N.P., and A.B.-K. interpreted data; and A.B.-K. and T.C. wrote the manuscript with contributions from all other authors.

## Competing interests

The authors declare no competing interests.

## Additional information

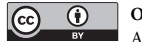

