## [Peer Review File · Nature Communications]

REVIEWERS' COMMENTS

Reviewer #1 (Remarks to the Author):

The authors have satisfactorily addressed the points raised in the original review. The manuscript is much improved. Only minor points remain.

1. In the abstract, the authors state that “the host phosphatidylglycerol biosynthesis processes appeared to have major anti-viral functions.” This seems premature because they have not experimentally validated that these processes indeed are antiviral let alone that these processes are “major”.
2. Fig 4E, 6d it says PFU on the graph but mentions q-rtPCR in the legend, please check.

Reviewer #2 (Remarks to the Author):

Upon reviewing the revisions to the submitted manuscript “CRISPR screens for host factors critical for infection by SARS-CoV-2 variants of concern identify GATA6 as a central modulator of ACE2” by Israeli et al., we find the authors have meticulously addressed all of the comments, not only from us, but from the other two reviewers as well. The authors have addressed our largest concern—that identification of differential host-factors between strains may have been overambitious given the noise of the assay—largely by removing conclusions that could not be readily validated. However, by performing additional experimental validation, the authors still demonstrate that dependence on host-factors can differ between viral strains even if a robust statistical analysis of the bulk CRISPR screen was not possible. Moreover, through the additional supplemental material (replicate correlation, and full CRISPR screen datasets), their results are more accessible and transparent.

Overall, we feel that even if a slight reduction in scope of the manuscript's claims has occurred, as a whole the manuscript is substantially improved by the revisions and the solid conclusions that have been left are compelling. This is even more the case in light of the numerous additional experimental results the authors have added to address the other reviewers' comments. We recommend the manuscript for publication. One minor comment that remained on our end: the supplemental figure 1 may be presented better if the density plots shown in the response letter can be included.

Reviewer #3 (Remarks to the Author):

The revised manuscript adequately addresses my and the other reviewers initial comments and I now find the revised manuscript suitable for publication.

Point-by-point letter of response to the reviewers

We would like to thank the reviewers and the editor for their time and valuable comments, which we feel helped us to substantially improve our manuscript. The study has been expanded and the manuscript has been amended extensively, addressing all comments.

Reviewers Comments:

Reviewer #1 (Remarks to the Author):

In this manuscript, Israeli et al employed genome-wide CRISPR screens in Calu-3 to identify host factors that are essential for the infection by WT-SARS-CoV-2 as well as two additional variants of concerns (VOCs), Alpha and Beta. The screens provided lists of candidate genes with pro-viral functions. The authors follow up on GATA6, a transcription factor that scored highly in screens with all three viral strains. The authors demonstrated that GATA6 promotes ACE2 transcription, likely through binding of the ACE2 promoter. Although the inclusion of the VOCs is interesting and a likely mechanism by which GATA6 acts is presented, the study is quite limited in scope. It focuses on a single host factor in a single cell line (Calu3), which makes it difficult to assess how it compares to cellular factors identified in numerous other manuscripts in the literature describing SARS-CoV-2 screens. Besides GATA6 none of the potentially interesting candidates, some of which may be specific for certain VOCs, are experimentally validated.

Major concerns

1) The described screens add to a growing list of candidate host factors identified through CRISPR screens. The authors have widely discussed multiple potential pro- and anti- viral genes that are common and distinct to the virus variants tested but basic validation of the candidates is lacking. The authors use Venn diagrams to look for common and specific factors but due to variation between screens one cannot conclude whether these differences are meaningful especially because there are no biological replicates. Experimental validation is required for some of the more novel proviral factors. Is KREMEN2 really specific for a certain VOC and are the other novel factors (e.g. IRF6 and CUL5) specific for Calu-3 or other cell types? In addition, the antiviral factors, such as RPP21, UQCRC1, and MRPS genes, are in general common essential genes for cell proliferation, it is important to know whether the significant depletion of these genes are due to the defects in cell growth or their anti-viral properties. The essentiality of these genes on the infectivity of the three virus variants inspected should be tested accordingly through gene knockdown or knockout using RNAi or CRISPR. In the absence of any validation, the list of genes is not very informative.

Additional experiments were performed to validate the results of our screens. As suggested we generated KO cell lines using CRISPR and probed the effect of disruption of individual candidate-genes on the infectivity of the virus (WT and VOCs). These experiments allowed us to confirm the results of our screens (Fig. 1E, 1F, and 3D).

Regarding the variation between screens, there is indeed, a considerable amount of noise in the system, however the top hits are highly reproducible. In response to the reviewer concern, in the amended version of the manuscript we incorporated a supplementary figure (Supp. Figure 1, reproduced below) depicting the correlation of gene enrichment between biological replicates, demonstrating reasonable reproducibility between the strongly affected genes.

Since the variability within the screens is high, we cannot identify differential dependencies between strains in a comprehensive manner. However, we were able to pinpoint a few promising genes which are differentially selected in the different strains and are consistent across replicates of screens carried out with the same strain, such as KREMEN2. In the revised manuscript, we also included additional data demonstrating experimental validation of these hits (Figures 1E,1F,1H and 3D), providing further support to the conclusions of the screen results.

Following the reviewer comment, we removed the comparison between the top 100 genes in each screen (originally Ven Diagram) since we agree that it is impossible to assess accurately how much this indeed reflect bona fide differences between strains. However, since comparing the host response to infection by different SARS-CoV-2 strains is of great value, we assumed that analysis using less strict thresholds, might allow us to “pick” meaningful candidates. Indeed, we identified KREMEN2 and SETDB1, which are consistently enriched in the Alpha strain screen, but are depleted from the other two screens. To illustrate that these genes represent true hits and not noise, we generated KREMEN2 and SETDB1 gene-disrupted cell-lines and examined their infection by the WT SARS-CoV-2 and two VOCs. The results confirmed the strain specificity of the pro-viral function of these genes (Fig. 3D).

Regarding the cell specificity of novel factors identified by our screens, inspection of the relative RNA expression in published mRNA-seq datasets of Calu-3, A549 and Vero-E6 cell lines, revealed that GATA6 and IRF6 are expressed at much higher levels in Calu-3 cells (IRF6 expression is below threshold in Vero-E6 and A549) (Fig. 1G). CUL5 expression appeared to exhibit similar levels in various cell-lines, therefore it is conceivable that its detection in the current screen, may be attributed to a Calu-3 cell-specific modulation in the course of infection. Taken together, the data therefore suggest that some of the host entry factors necessary for SARS-CoV-2 infection exhibit cell-line specificity, illustrating the importance of using relevant cell models to better understand SARS-CoV-2 entry. These data are now included in the revised manuscript.

The reviewer is correct in pointing out that since gene KOs may affect cell viability as well as infection outcome, the gene depletion results from the screens may be difficult to interpret. To improve the analysis, we performed an additional control “viability” screen in which the cells were allowed to grow for 7 days, at the end of which, those containing sgRNA that affect viability are expected to be depleted. We compared the sgRNA population to that of day 0, and performed gene set enrichment analysis the same way we did for the infection screen. The results show that most pathways depleted in the infection screen were also depleted from the viability screen. The only functional pathway uniquely depleted upon infection is Phosphatidylglycerol biosynthetic process and this is now extensively discussed (see novel Figure 2A). In light of the new analysis, our ability to capture depleted genes is rather limited and we therefore focused mainly on the enriched genes.

2) GATA-6 has not been identified in other CRISPR screens performed in different cell types. The authors claim that this indicates that it is a cell type specific host factor but do not provide any supporting experimental data. Moreover, GATA6 is also not in the top hits in Calu-3/SARS-CoV-2 screens done by two independent groups (ref (21) and PMID: 34075371). At minimum the authors should knockout GATA6 out in other cell types to provide evidence of cell-type specificity.

Inspection of published data revealed that GATA6 and many of the pro-viral genes identified in our screens were also enriched, to some extent, in other Calu-3/SARS-CoV-2 surveys (Fig. 1H, see also below).

In addition, as suggested by the reviewer, a GATA6 gene disrupted cell-line was generated in the background of Vero-E6 cells. In this cell-line the infectivity of the virus was not affected substantiating the importance of conducting the screen in human epithelial cells (Fig. 4F,4G and Supp. Figure 2).

3) The authors showed virus-induced CPE and virus titer were significantly reduced in GATA6-disrupted cells, could the author add back the GATA6 cDNA and show the reversed phenotype? Curiously, this important control experiment is described in the material and methods, but not presented in the results.

Trans-complementation experiments in which GATA6 was expressed in the KO cells were conducted and resulted in restoration of the phenotype (Fig. 5B, Fig 5D, and Fig .5E).

4) The authors suggest that GATA6 is involved in the early virus entry step. They present reduced immunofluorescent signals against SARS-CoV-2 in GATA6-disrupted cells at 48hpi as data supporting this. However, at this late time point, the reduced virus infection could be due to defective virus entry, replication, and/or translation. Stronger evidence is from severely reduced ACE2 levels, which given the well-established role of the SARS-CoV-2 receptor ACE2 in viral entry, links GATA6 to viral entry. However, knockout of the transcription factor GATA6 likely results in widespread changes to the transcriptome, which effects reach beyond ACE2. Further experimental evidence is needed to conclude that GATA6 is related to virus entry or that reduced ACE2 levels fully explain the phenotype. Besides, the highlight of this manuscript is to identify host factors critical for the infection of SARS-CoV-2 variants, to validate the antiviral effect of GATA6 with the other SARS-CoV-2 variants may be necessary to emphasize the significance.

The original manuscript included data which inadvertently was attributed to a 48 hrs time-point. Actually, the data pertained to samples examined 24 hours post-infection. We now added data from 5 hrs. post-infection, further supporting the conclusion that GATA6 exerts its function at the phase of the viral entry into the host cells (Fig. 5A). It is important to note that we also were successful in restoring the phenotype of the GATA6 KO cells by trans-complementation with ACE2. This result represents a strong genetic support for the conclusion that the effect of GATA6 on infection involves modulation of the ACE2 receptor. These new experiments are included in the revised manuscript (Fig. 5D, and 5E) and the text modified accordingly.

6) In this manuscript pyrrothiogatain is used as a GATA6 inhibitor. In reference [71] that describes the discovery of pyrrothiogatain it was experimentally shown that it inhibits the in vitro DNA-binding activity of the GATA family of proteins including GATA2,3,4 and 5. GATA1 and GATA6 were excluded due to technical reasons. At high pyrrothiogatain concentrations, the authors see a modest decrease in GATA6 expressions levels, which the authors interpret as evidence that pyrrothiogatain also inhibits GATA6. This is confusing because in reference (71), the authors show that pyrrothiogatain inhibits DNA binding activity but that expression levels (they looked at GATA3) are unaffected. From the data presented it cannot be concluded that pyrrothiogatain (1) inhibits GATA6, (2) the concentration required for antiviral activity corresponds to concentration required for GATA inhibition, and (3) the antiviral effect is not due to inhibition of other GATA members or due to off-target effects. Moreover, the data regarding the antiviral effect is shown at just one concentration, is not quantified, and no replicates are shown to demonstrate statistical significance. It is unclear what the IC50 is for replication inhibition and although there is mention in the text that pyrrothiogatain is not cytotoxic, this data is not included in the manuscript.

In line with the reviewer comment, we conducted EMSA in the presence of increasing concentrations of Pyrrothiogatain and demonstrated that the drug lowers the DNA binding of GATA6 to its target ACE-2 promoter DNA sequence. Regarding the inhibition of the viral infection, the Pyrrothiogatain concentration selected was that found to affect the binding of GATA6 to the ACE2 promoter and to lower the level of ACE2 transcription. This concentration is not toxic to the cells. This information is now included in the manuscript (Fig. 6A and 6B). New data included in the revised manuscript was obtained from experiments carried-out in triplicates and statistical significance is demonstrated (Fig 6D). The data cannot rule out the possibility that other members of the GATA family are affected by Pyrrothiogatain, yet other previous transcriptomic surveys did not evidence the upregulation of other GATAs neither did our screen distinguish the enrichment of other GATAs upon infection (Table 1). Furthermore, with the exception of GATA6, none of the other 5 members of the GATA family revealed significant levels of transcription in the Vero-E6, A549 and Calu-3 cell lines. These data are now included in the revised manuscript (Fig. 6F).

(7) Throughout the manuscript there is a lack of statistical testing whether the reported differences are statistically significant. For examples figures 4B,4D, 5C, 5D, 5H, S1, and S2.

Statistical analyses were conducted for all experiments in the manuscript, as detailed in the revised respective legends to figures.

Minor comments

1) It is a bit confusing that first the combined screening results of all three viruses is presented followed by splitting them up in their individual parts. This could be described a bit more explicitly because e.g. "Figure 1. CRISPR genome-wide screens in human Calu-3 cells identify host factors important for infection by SARS-CoV-2." might suggest that this presents results only from wild-type SARS-CoV-2.

The combined and individual data set analyses is further elaborated in the text of the revised manuscript and in the relevant legends to figures, as suggested (Lines 151-154)

2) The resources and the methods to obtain the relative expression levels of GATA6 in different cell lines is not clearly described in the result part. Figure 4C and 4D are not clearly annotated.

The requested information was incorporated in the revised manuscript and legends as requested (Lines 233-241, 697-700, Fig 1G, and Fig 4I).

3) The authors showed the up-regulated GATA6 expression in virus-infected cells and clinical samples and considered GATA6 is involved in the susceptibility to viral infection. Will the overexpression of GATA6 in WT-Calu3 cells increase the infection?

We performed the suggested experiment and the data describing the change in ACE2 and viral infectivity following overexpression of GATA6 are presented as supplementary Figure 3. The results show that the level of ACE2 indeed increased, yet no change in infectivity was detected.

4) Bar chart figures should include the side scale for y-axis and error bar. Annotation for qPCR should clarify the control gene to make the figures self-explanatory.

Done as requested.

5) Figure 5A and 5H, the green signal is the staining using antisera against SARS-CoV-2, not GFP signal. The label should be corrected accordingly.

Done as requested.

6) Figure 5B: include the western blot against GATA6 to show its KD level.

Done as requested.

7) The authors should provide the results of the screens in a spreadsheet format.

Done as requested (Table 1-3).

Reviewer #2 (Remarks to the Author):

In this manuscript Israeli et al perform a genome-wide CRISPR knockout screen to identify human genes linked to susceptibility to or protection from SARS-CoV-2 infection. The study is conducted in a human lung epithelial cell line, Calu-3, and the authors provide a compelling argument that using a cell line similar to targeted host cells is important for identifying the most clinically relevant host factors. The study uncovers novel host factors linked to infection and validates several that have previously been reported. By repeating these experiments in the wildtype SARS-CoV-2 strain as well as two recent variants of concern (Alpha and Beta), the authors suggest new relevant targets specific to infection by different strains. Most importantly, the authors identify GATA6 as a novel host factor essential to infection from all three lines, and show that GATA6 has much higher expression levels in Calu-3 cells, which was why it was missed by previous CRISPR screens. Through follow-up experiments, the authors confirm GATA6 is a key transcription factor regulating ACE2 expression (thus providing a clear mechanistic explanation for its role in infection), suggest that GATA6 is specifically upregulated by SARS-CoV-2 infection, and verify that pharmacological inhibition of GATA6 is capable of protecting cells from infection.

Overall, we are enthusiastic about this manuscript. Given the validation that identified pro-viral genes can be viable drug targets, we believe this work will be of considerable interest to the community. Nonetheless, we have several major comments that if addressed we feel would strengthen the manuscript and improve confidence in the analyses. In particular, though the validation of GATA6 is convincing, interpretation of the less extreme hits could be extended, and we feel more thorough quality assessment may be necessary for the CRISPR screen as a whole (especially in regards to comparing results between viral strains).

MAJOR COMMENTS

1) Most importantly, we find that further quality assessment of the CRISPR screen is necessary to instill confidence in the results. In particular because of the poor overlap and low correlation for the enrichment scores between viral strains, we are concerned about the overall reproducibility of the screen. The authors only include two replicates for each screen. Is this sufficient for robust statistical analysis? To confirm whether high reproducibility could be achieved between replicates, supplemental figures showing the enrichment score correlations (similar to figure 3.C) should be provided. If the assay as a whole is poorly reproducible, the top hits may still be reliable, but meaningful comparison between strains may be impossible.

As the reviewer noted, there is a considerable amount of noise in the system, however the top hits are highly reproducible. In response to the reviewer concern, in the amended version of the manuscript we incorporated a supplementary figure (Supp. Figure 1, reproduced below) depicting the correlation of gene enrichment between biological replicates, demonstrating reasonable reproducibility between the strongly affected genes.

Since the variability within the screens is high, we cannot identify differential dependencies between strains in a comprehensive manner. However, we were able to pinpoint a few promising genes which are differentially selected in the different strains and are consistent across replicates of screens carried out with the same strain, such as KREMEN2. This is now documented in an additional figure (Supp. Figure 1, reproduced below).

In the revised manuscript, we also included additional data demonstrating experimental validation of these hits (Figures 1E,1F,1H and 3D), providing further support to the conclusions of the screen results.

2) Related, the authors note that agreement between individual guides targeting the same genes (figure 1.B) is indicative of high technical quality of the data. We note that the range of the x-axis suggests some guides are enriched at least 2-fold more than the most enriched GATA6 guide (i.e. scaled directed score of 40 compared to 20). Were these highly enriched guides also significant at a gene level? If not this large range could indicate a high degree of noise in the assay at an individual guide level. The authors should add a track containing the 1,000 non-targeting control guides to provide an estimate of the range of scaled directed scores observed purely from noise in the assay.

We thank the reviewers for bringing this to our attention. There was, in fact, an error in the figure, which led to unnecessary and incorrect stretching of the x-axis. After correcting this error, it is apparent that GATA6 and ACE2 have high sgRNAs scores relative to the population, and there are no sgRNAs that 2-fold more enriched. In fact, the top two scoring sgRNAs target GATA6 and ACE2. For the sake of clarity, a table summarizing the sgRNA level results was added (Table 2). As suggested by the reviewers, a track for the non-targeting control sgRNA has been added to figure 1B.

3) The clear definition of significant hits is elusive throughout the paper, and confounded by the presentation of the figures. When and why individual points are labeled and / or colored (e.g. Figure 1.C, 1.D, 3.A, and 3.C) is not always clear. Additionally, the analyses for gene set enrichment analysis and overlap between strains are inconsistent between whether a specific sub-set of "hits" is used or if the top X genes are selected. For improved transparency and further analysis of the results, the authors should provide a clear list of hits alongside supplemental tables including the raw enrichment scores and raw / FDR adjusted p-values for all guides or genes.

As suggested by the reviewer, we now included supplementary tables summarizing the results at the gene and sgRNA levels (Table1-3). The tables include fold-change and FDR values for statistical significance assessment.

The gene set enrichment analysis was performed on the ranking of all the genes in the screen, and the overlap comparison was removed from the manuscript. Figure legends were rephrased throughout the revised manuscript for better clarity.

4) Although the comparison of essential host-factors for viral infection with each strain would be of considerable biological interest, we do not find the current results believable. The results presented (Figure 3.B and 3.C) suggest considerable differences in essential host-factors between strains (despite the reported only 17 nonsynonymous differences between Alpha and WT). The correlation between strains just is not good and it is unclear if this is because the biology of infection truly is that different between strains or because the assay itself is too noisy for valid comparison (see point 1). Moreover, the overlap analysis focuses on the top 100 most enriched genes for each strain. We are not convinced this is this a reasonable presentation strategy. If in fact there are only a handful of genes that are true pro-viral factors (e.g. Figure 1.C only shows about 10 genes at $FDR < 0.05$ ($-\log_{10}(FDR) > \sim 1.3$)), comparison of the top 100 "most enriched" (but mainly insignificant) genes may only be comparing fluctuations due to noise. In fact, from an $FDR < 0.05$ call on 1.C it would appear only ACE2 and GATA6 are significantly enriched pro-viral genes. In other words, comparing only significant hits, the authors may find perfect overlap between the three strains; the exact opposite of the interpretation presented by the authors. The authors should further clarify if all of the top 100 pro-viral genes are considered significant hits and justify their inclusion in this comparison if they are not.

We agree with the reviewer comments regarding significant noise in our screens, which limits the ability to compare between strains. As discussed above (point 1), although the top hits are consistently enriched throughout all screens, the overall correlation within and between the screens is not strong. Following the reviewer comment, we removed the comparison between the top 100 genes in each screen (originally figure 3B) since we agree that it is impossible to assess accurately the fraction of the genes that reflect bona fide differences between strains.

However, since comparing the host response to infection by different SARS-CoV-2 strains is of great value, we explored if analysis under less strict thresholds, will allow to "pick" meaningful candidates. By doing so, we identified KREMEN2 and SETDB1, which are consistently enriched in the Alpha strain screen, but are depleted from the other two screens. To support that these genes represent true hits and not noise, we now confirmed the strain specificity of the pro-viral function of these genes by infection of gene-disrupted cell-lines (Figure 3D).

5) We further note that although the pro-viral hits are discussed thoroughly, comparably little analysis was done on the anti-viral hits. While we understand this may be in part motivated by the fact that the pro-viral genes are the best candidates for informing treatment options, we find this decision odd considering the data. In particular, the anti-viral genes seem to be more significant overall (1.C). Moreover, comparing viral strains, the reproducibility and correlation appears to be considerably better among the depleted (anti-viral) genes. In fact, by eye (and by labels in 3.C) it would appear that an overlap comparison similar to 3.B would show much better agreement for anti-viral host factors between strains, but this is never discussed. The authors should consider further discussion and analysis of the anti-viral genes and their importance in understanding SARS-CoV-2.

The design of the screen relied on cell viability as a readout for productive infection. Since gene KOs may affect cell viability as well as infection outcome, the gene depletion results from the screens were difficult to interpret. To improve this analysis, we performed an additional control screen in which the cells were allowed to grow for 7 days, at the end of which, sgRNA that reduce viability are expected to be depleted. We compared the sgRNA population to that of day 0, and performed gene set enrichment analysis the same way we did for the infection screen. The results show that most pathways depleted in the infection screen were also depleted from the viability screen. The only functional pathway uniquely depleted upon infection is Phosphatidylglycerol biosynthetic process and this is now discussed in the revised manuscript (see novel Figure 2A). In light of the new analysis, our ability to capture depleted genes is rather limited and we therefore focused mainly on the enriched genes.

6) For the comparison between GATA6 expression in infected vs. uninfected individuals, the insignificant change in expression for ACE2 should also be shown as this negative result is relevant to the authors' interpretation (i.e. they posit that elevated GATA6 has some role in infection not directly linked to ACE2 expression).

In response to the reviewer comment, the revised manuscript includes additional data pertaining to the level of ACE2 in clinical samples (Fig. 4H).

7) Finally, the results showing Pyrrothiogatain treatment protects against viral infection is particularly interesting. However, we are curious if transcription factors generally make good drug targets for clinical application. If GATA6 regulates expression of many genes would there be considerable concern about unintended side effects? The lack of toxicity noted is encouraging, and we understand the authors do not explicitly suggest Pyrrothiogatain should be employed for therapeutic, but some discussion of past clinical successes from drugs targeting transcription factors could better contextual the potential impact and utility of the identified pro-viral hits.

The reviewer is correct in pointing out that transcription factors are not ideal therapeutic targets, yet recently such factors (including GATA family members) became increasingly relevant for therapeutic purposes. A paragraph discussing this paradigm shift is included in the revised manuscript (Lines 514-527).

MINOR COMMENTS

1) we additionally had several minor comments, questions, or points of interest. We do not necessarily believe all of these need to be addressed, but include them for thoroughness and in case the authors may find any of them useful. 1) The results would be more impactful if the Gamma and Delta variants could have been included in the study, however we understand these strains may not have been available at the inception of the experiment.

As stated by the reviewer, only the Alpha and Beta variants were available when we performed the screens. Yet we included in the revised manuscript data obtained in experiments in which GATA6-KO cells were infected with the widely spread Delta variant, yielding similar results (Fig 4D and 4E). Unfortunately, the availability of the Delta variant or additional variants for further experimentation in our lab is limited.

2) We found it interesting that this screen only identified GATA6 despite the fact that other transcription factors (including GATA1 and GATA3) are reported to target ACE2 (<http://bioinfo.life.hust.edu.cn/hTFtarget#!/targets/gene?gene=ENSG00000130234>). The authors may consider adding discussion as to why other transcription factors (especially GATA family TFs) were not enriched (e.g. cell type specific?) and / or highlighting if any showed signs of being weaker pro-viral hits.

In response to this comment, additional analyses were carried out to determine the expression level of GATA family members in CALU3 and other cell lines. The data establish that indeed GATA6 is expressed at considerable higher levels. Furthermore, RNA-seq data established that upon infection of Calu-3 cells no induction of other GATA members is detected. These results are now documented in Fig. 6F and discussed accordingly in the text.

3) We find some of the results presenting the enrichment / depletion of hits to be confusing. In particular, Figure 1 contains 3 different quantitative scores (scaled directed score (1.B), log fold-change (1.C) and enrichment / depletion scores (-log₁₀) (1.D). It's difficult to contextualize the differences between these scores or how to directly interpret the magnitude of their effect sizes. Similarly, in several figures (1.C, 1.D, 3.A, and 3.C) it is unclear why individual points have been labeled. For 1.C many points that appear to not reach an FDR < 0.05 are labeled. Are the points only labeled because they are discussed and not necessarily significant? Are there other genes that were significant but not labeled? In 3.A, which genes are labeled appears arbitrary. For instance, some genes (e.g. KREMEN2 or ARIH2) are highlighted in all three strains despite only being enriched in some, but other

genes (e.g. AP1G1) are similarly only enriched in one strain, but only labeled in that one strain. Further, the interpretation of the size of the points in 1,D and 3.A is not clearly explained.

Figure 1 was revised in response to the reviewer concern. It now contains three standard presentations of CRISPR screen results. Panel B shows sgRNA level results using a score for the sgRNAs. Panel C and panel D contain gene level results, panel C being a gene level volcano plot that displays the fold change of sgRNAs on one axis and the statistical significance on the other, allowing evaluation of the overall outcome of the screen. Panel D shows MAGeCK RRA enrichment scores. These scores are calculated for each gene based of the rank distribution of the sgRNAs targeting it. We chose to include two different presentation for the gene results since it is a central theme in the manuscript. The text includes now explicit reference to the genes which are marked in panels C and D of Figure 1.

4) The results from Figure 1.B and 1.C clearly show an asymmetry between enrichment and depletion (particularly in level of significance in 1.C). Would it be helpful or possible to consider enriched vs. depleted genes separately for directed score scaling and / or multiple hypothesis correction?

The negative effects of gene KOs on cell viability as well as infection outcome are difficult to distinguish in our screen. We therefore focus more on the enriched genes of the screen results, and less on the depleted genes (see also response to comment 5).

5) For the gene set enrichment analysis in Figure 2, the authors state that the analysis was performed on hits from all strain screens combined. It is unclear if this is a good idea considering the authors' claim that different strains depend on almost wholly different host-factors. Moreover, it is unclear exactly how this gene set enrichment analysis was performed. The methods claims that directed scores were used as input for gene set enrichment analysis. Did the analysis perform a ranked GSEA using the directed scores from all genes (using the maximum, minimum, or mean score from all three strains?), or were the results filtered to a set of "hits" within the three strains?

As stated above, only a handful of genes were significantly enriched in the screens. Accordingly, the gene set enrichment analysis was focused on genes depleted in the screens. We have added a more detailed explanation of the GSEA in the methods that clarifies the exact analysis. The new text reads as follows: "Directed scores calculated from MAGeCK RRA scores of all screens combined (treating all screens as one experiment) were used as input for gene set enrichment analysis (GSEA version 4.1) with GO biological process (c5.bp) from MSigDB (version 7.4)".

6) We additionally note two artifacts in the data in Figure 3. There appears to be a large gap in enrichment scores just under 1 for the Beta strain. Additionally, many genes have near zero enrichment score in the Alpha strain despite very large scores in the WT or Beta strain

(large vertical density in first and second plot in 3.C). We are curious if there is any interpretation of these artifacts?

We thank the reviewers for pointing out these artifacts. It is unclear to us why there is a gap in enrichment scores in the Beta screen. The Alpha screen seemed to have weaker selection relative to the others, and therefore many genes have weak fold-change surrounding zero LFC. An alternative explanation can be contamination of the original plasmid in the sequencing of the Alpha strain results. In our interpretation of the results, we address the extreme changing genes, which are not affected by the above artifacts. Therefore, this does not affect the overall conclusions.

7) The plot for gene set enrichment analysis (2.A) could perhaps be presented more clearly? The shape and coloring of the scatterplot seem to indicate a meaningful separation. In other words, from a brief look, one might take away that Mitochondrial Translation was much more significant than Respiratory Electron Transport Chain. But on closer inspection this does not appear to be the case, as the x-axis for enrichment score only spans from -2 to -1.9 and the coloring for FDR adjusted p-value spans from 0.002 to 0.008.

The original Figure was omitted (see also response to comment 5). The following information was added to the legend of the novel panel (2A): the enrichment of each pathway in each screen is represented as bubbles colored by Normalized enrichment score (NES) and sized by significance levels.

8) The error bars in 4.B may be misdrawn (we expect these to show mean +/- SE, but they are not centered on the mean. Perhaps they are currently only showing mean + SE)? Additionally, average result from all replicates and proper error bars should be provided for 4.E as well as 5.C and 5.D. Depending on the setup for this data it may not make sense to show error bars on the control, but estimates of the error should at least be shown for the comparisons.

Statistical analyses were conducted and added for all experiments in the manuscript, as detailed in the respective legends to figures.

9) The figure legend for 4.D is mislabeled and mistakenly indicates that the individual values are shown for 5 hours post infection instead of 8 hours post infection.

The legend had been changed as follows: Values from single replicates shown for 03hr and 05hr. Mean values of duplicates shown for uninfected and 08hr in red, and individual replicate values are presented as black points.

10) It may be important to further clarify whether the nasal swabs from 4.E were from individuals who tested positive / negative or individuals who were symptomatic / asymptomatic. The interpretation the authors propose is that GATA6 expression was higher

in infected individuals (and presumably elevated as a result of infection). It may also be possible that individuals with naturally higher GATA6 expression were more susceptible to infection to begin with. Additionally, it could be possible that variations in GATA6 expression impact the trajectory of infection; individuals with lower GATA6 expression may be more likely to be asymptomatic even if they are infected?

We have no information pertaining to the extent of symptoms manifested by each positive individual tested. An explicit statement was incorporated in the revised Materials and Method section (line 706). While we cannot rule out the possibility that natural variations exist in the levels of GATA6 expressed by different individuals, the increase in the transcription level of GATA6 observed upon infection (Figure 4I), suggests that the GATA6 modulation is not a cause but rather an effect.

11) We additionally suggest that the result showing increased GATA6 expression (4.D) in infected cell lines over time may be more compelling if it is presented after the data from nasal swabs (4.E) as a confirmation that infection itself increases expression. At the very least, discussion of these two results should be done together. Currently 4.D is mentioned alongside the results comparing expression between cell lines from 4.C, but we do not see any clear link for the 4.D data in this argument (that it is conceivable the function of GATA6 is not fulfilled identically in A549 and Vero-E6 cell lines). There is however a direct connection between the results in 4.D and 4.E and what they suggest about GATA6's role / regulation in infection.

The suggestion of the reviewer was highly appreciated and accordingly, the order of the presented data was changed.

12) It would have been interesting to explore direct independent validation of the pro- and / or anti-viral effects of some of the less significant hits outside of GATA6 and ACE2 (if there are any) similar to 4.A and 4.B.

Validation experiments were carried using gene-disrupted cell line and the results were included in the revised manuscript (See Figure 1E).

13) Knockout of GATA6 seems to have had at least as extreme (possibly slightly stronger) an effect on ACE2 expression as knockout of ACE2 itself (5.C). This is despite the fact that knockout of GATA6 may have been incomplete (only 60% control levels). Is this result surprising and / or worthy of note?

This difference is not significant, as shown in the novel Figure 5C.

14) Can the authors explain the decrease in GATA6 expression following treatment by Pyrrothiogatain? If Pyrrothiogatain merely inhibits the DNA binding function of GATA6 and other GATA family members, it may be unintuitive that the protein levels of GATA6 decrease following treatment. Can this clearly be explained (e.g. by Pyrrothiogatain induces a

conformational change in GATA6, or perhaps an expression feedback loop involving GATA6 regulating itself)?

New EMSA results were included in Figure 6A showing that Pyrrothiogatain inhibited the DNA binding of GATA6 to its consensus sequence. The possibility that a conformational change of GATA6 is induced by the drug and/or the GATA6 protein level decrease involves a feed-back mechanism, are discussed in the text of the revised manuscript (Line 497).

15) We noted a couple of small typos throughout the manuscript (e.g. "SRAS-CoV-2 receptors" (page 9) and "mRNA evels" in the Figure 4 legend).

All typos were corrected.

Shayne Wierbowski and Haiyuan Yu

Reviewer #3 (Remarks to the Author):

While this manuscript touches on a topic of interest in the development of novel broad-spectrum host-directed therapeutics through the identification of critical cellular factors that are exploited by SARS-CoV-2 infection. In my opinion, the manuscript might be suitable for publication in Nature Microbiology, after the authors have addressed the following comments and questions:

1) In the abstract and the rest of the paper, the authors should refrain from referring to the KREMEN2 receptor as a "unique" gene required only by the Alpha variant as they do not further validate this claim beyond the screening and "may be a unique" gene would be more appropriate in this content. The screening in this paper further fails to demonstrate the identification of KRMEM1, which has been published.

The involvement of KREMEN2 was further validated by gene-disruption followed by infection with the various VOCs. This new experimental data supports our original statements and is now included in the revised version of the manuscript (Figure 3D). In response to the reviewer comment, the phrasing was modified.

2) "Remarkedly, in a recent screen in search of additional SRAS-CoV-2 receptors, KREMEN1 was identified as an alternative viral-entry receptor [63]." Authors misspelled SARS here.

The inadvertent mistake was corrected.

3) In Fig.4A, the authors should make it clear how they are able to observe cytopathic effects in Calu-3 cells infected with SARS-CoV-2 as obvious cytopathic effects are not seen in this cell-line. The authors should also include a cell viability assay for WT, GATA6 KO and ACE2

KO mock and infected cells in Fig.4. It is also not clear why the MOIs and 48 hours post infection were chosen. The authors should perform a growth curve/time course of infection here. Viral genome copy numbers could be added to complement Fig.4B and western blot analysis showing protein levels of GATA6 in the different cell lines would be informative (Fig.4C and D).

While this effect was not reported by all laboratories, in our and others hands cytopathic effect can be detected in infected Calu-3 cells (see below).

Cell viability controls were performed using crystal violet staining and are included in Fig. 4D and 4E. As suggested by the reviewer, the time-course experiment was performed (Fig. 4B). Genome copy numbers were inserted in the relevant figures through-out the revised manuscript. We performed Western blot analyses on other cell-lines (A549 and Vero-E6) yet unfortunately, using available antibodies. GATA-6 was under the detection level

4) The authors should mention that the upregulation of GATA6 in nasopharyngeal swab samples has already been published "Transcriptome of nasopharyngeal samples from COVID-19 patients and a comparative analysis with other SARS-CoV-2 infection models reveal disparate host responses against SARS-CoV-2".

Thank-you for bringing this report to our attention. This information was incorporated in the revised manuscript (Lines 416).

5) The authors should fix the typo in Fig.4E legend "levels" instead of "evels".

All typos were corrected.

5) In Fig.5 the authors should make it clear what MOIs they are using. The authors should also include the immunofluorescence staining of GATA6 and ACE2 in mock and infected WT, GATA6 KO and ACE2 KO cells. Fig5B should also include a western analysis of GATA6.

In the revised manuscript, the MOI information was added to the legends of the relevant figures. As suggested, the levels of expression of GATA6 and ACE2 were analyzed by Western-blot and RT-PCR. The data enabled accurate quantification of the expression level (Figure 5B, 5C). Immunofluorescence studies were not carried-out due to technical reasons.

6) IF staining showing the (co)localization of GATA6 and ACE2 in mock and infected cells should be added. Does the SARS-CoVs S1 or RBD of spike protein bind or colocalizes with GATA6? Overall, the validation of GATA6 as a requirement for infection by SARS-CoV-2 were not sufficiently explored. Why the Alpha and Beta variants are not included in the validation is a mystery. Furthermore, the analysis of GATA6 in clinical samples has already been shown (see above comment for the article) and the role of GATA6 in entry is inferred in this study through "modulation" of ACE2 expression and should be adequately study for suitability to publish in this journal.

Our data suggested that the transcription factor GATA6 promotes ACE2 transcription through binding to its consensus sequence which is present in the ACE2 promoter region. Since transcription factors are localized in the nucleus, we do not expect colocalization of the transcription factor GATA6 and the ACE2 receptor or to the SARS-CoV-2 spike protein. The involvement of GATA6 was further validated by the following experiments: (1) Infection of GATA6 KO cells was now also examined at short time post infection (5 hrs) further supporting the role of GATA6 in the entry phase; (2) Wild type SARS-CoV-2 viral load was quantified along various times post infection of WT and GATA6 KO cells; (3) we performed infectivity analyses (by imaging, quantification of virus and cell viability) with the WT virus and other VOCs. (4) we performed trans-complementation experiment with ACE2 in GATA6 ko cells which restore the phenotype of the cells. This result represents a strong genetic support for the conclusion that the effect of GATA6 on infection involves modulation of the ACE2 receptor. All of these new experiments are now included in the revised manuscript (Figs 5A,4B,4C,4D,4E,5D,5E) and the text modified accordingly.

Reviewers Comments after revision:

Reviewer #1 (Remarks to the Author):

The authors have satisfactorily addressed the points raised in the original review. The manuscript is much improved. Only minor points remain.

1. In the abstract, the authors state that "the host phosphatidylglycerol biosynthesis processes

appeared to have major anti-viral functions.” This seems premature because they have not experimentally validated that these processes indeed are antiviral let alone that these processes are “major”. In the revised abstract the sentence was omitted.

2. Fig 4E, 6d it says PFU on the graph but mentions q-rtPCR in the legend, please check. A PFU calibration curve tested in parallel was utilized to express Ct values as calculated PFUs. This information was added to the Methods section of the manuscript.

Reviewer #2 (Remarks to the Author):

Upon reviewing the revisions to the submitted manuscript “CRISPR screens for host factors critical for infection by SARS-CoV-2 variants of concern identify GATA6 as a central modulator of ACE2” by Israeli et al., we find the authors have meticulously addressed all of the comments, not only from us, but from the other two reviewers as well. The authors have addressed our largest concern—that identification of differential host-factors between strains may have been overambitious given the noise of the assay—largely by removing conclusions that could not be readily validated. However, by performing additional experimental validation, the authors still demonstrate that dependence on host-factors can differ between viral strains even if a robust statistical analysis of the bulk CRISPR screen was not possible. Moreover, through the additional supplemental material (replicate correlation, and full CRISPR screen datasets), their results are more accessible and transparent.

Overall, we feel that even if a slight reduction in scope of the manuscript’s claims has occurred, as a whole the manuscript is substantially improved by the revisions and the solid conclusions that have been left are compelling. This is even more the case in light of the numerous additional experimental results the authors have added to address the other reviewers’ comments. We recommend the manuscript for publication. One minor comment that remained on our end: the supplemental figure 1 may be presented better if the density plots shown in the response letter can be included. Supplementary figure 1 was revised according to the reviewer suggestion.

Reviewer #3 (Remarks to the Author):

The revised manuscript adequately addresses my and the other reviewers’ initial comments and I now find the revised manuscript suitable for publication.